# Generalized Discrete Diffusion with Self-Correction

Linxuan Wang [* 1]   Ziyi Wang [* 1]   Yikun Bai [* 1]   Wei Deng [2]   Guang Lin [† 1]   Qifan Song [† 1]

## Abstract

Self-correction is an effective technique for maintaining parallel sampling in discrete diffusion models with minimal performance degradation. Prior work has explored self-correction at inference time or during post-training; however, such approaches often suffer from limited generalization and may impair reasoning. GIDD (von Rütte et al., 2025) pioneers pretraining-based self-correction via a multi-step BERT-style uniform-absorbing objective. However, GIDD relies on a continuous interpolation-based pipeline with opaque interactions between uniform transitions and absorbing masks, which complicates hyperparameter tuning and hinders practical performance. In this work, we propose a **S**elf-**C**orrecting **D**iscrete **D**iffusion (SCDD) model to reformulate pretrained self-correction with explicit state transitions and learn directly in discrete time. Our framework also simplifies the training noise schedule, eliminates a redundant remasking step, and relies exclusively on uniform transitions to learn self-correction. Experiments at GPT-2 scale show that our method enables more efficient parallel decoding while preserving generation quality. Our code is available at: `https://github.com/laaaarrywang/Self-Correcting-Discrete-Diffusion.git`.

## 1. Introduction

Large language models (LLMs) (Radford et al., 2018; 2019; Brown et al., 2020) have revolutionized the realization of intelligence and significantly enhanced productivity across domains, including software development (Guo et al., 2025; Gong et al., 2025) and content generation (Shao et al., 2024). LLMs adopt an autoregressive (AR) generation pipeline to predict the next token sequentially. Despite its simplicity, next-token generation becomes quite time-consuming for long sequences. Alternatively, motivated by the success of BERT (Devlin et al., 2019), masked diffusion language models (MDLMs) (Austin et al., 2021; Shi et al., 2024; Ou et al., 2025; Sahoo et al., 2024; Gat et al., 2024; Campbell et al., 2022; Nie et al., 2025; Ye et al., 2025) propose a probabilistic, order-agnostic generation paradigm, which potentially leads to much reduced inference latency due to parallel generation and boosts the development of reasoning (Song et al., 2025; Khanna et al., 2025) and agentic artificial intelligence (AI).

Despite their potential for fast parallel generation, mainstream MDLMs typically achieve only comparable performance to AR by decoding a limited number of tokens per generation step using non-trivial sampling techniques. Decoding more tokens often disrupts inherent token dependencies and degrades reasoning performance. To address this issue, various acceleration strategies have been proposed, such as semi-autoregressive generation with block diffusion (Arriola et al., 2025), block-level KV cache (Ma et al., 2025; Wu et al., 2025), and distillation-based models (Chen et al., 2025a;b; Qian et al., 2026; Zheng et al., 2026). However, these approaches inevitably increase model complexity, making them harder to maintain and scale.

To avoid these complexities, we focus on the simple and effective self-correction techniques for maximizing the parallel generation abilities (Jiang et al., 2025). A notable example is the confidence- or entropy-based *remasking* step used during parallel generation to rectify erroneous tokens and revise previous mistakes (Nie et al., 2025; Wang et al., 2026; Song et al., 2025; Chen et al., 2025b). Rather than relying solely on the inherent generation ability of MDLMs, Kim et al. (2025) further enhances self-correction by fine-tuning pre-trained MDLMs with minimal effort. However, due to the limited generalization ability of post-training (Ouyang et al., 2022; Sutskever & Patel, 2025; Yue et al., 2025), self-correction remains underexplored.

To improve the generalization of self-corrections, GIDD (von Rütte et al., 2025) studied the pretraining-based self-correction via a multi-step BERT-style objective that incorporates additional uniform transitions. However, its interpolation-based pipeline introduces opaque interactions between uniform transitions and absorbing masks, compli-

---

[*]Equal contribution  [†]Equal advising  [1]Purdue University  [2]Morgan Stanley. Correspondence to: Wei Deng <wei-deng056@gmail.com>.

*Proceedings of the 43rd International Conference on Machine Learning*, Seoul, South Korea. PMLR 306, 2026. Copyright 2026 by the author(s).

**Denoising Step 127:**

...PORTLAND -- The Lions `haveulhu` a report of several charges against the Argonauts, one of whom appears to be the CFL club's seventh deaf head `coachicultural` General manager Pat Moore confirmed in an `COMM` to ESPN's Scott...

**Denoising Step 128:**

...PORTLAND -- The Lions `have received` a report of several charges against the Argonauts, one of whom appears to be the CFL club's seventh deaf head `coach.` General manager Pat Moore confirmed in an `email` to ESPN's Scott...

*Figure 1.* Example of self-correction between two consecutive denoising steps (127→128). Generated by SCDD ($p_u = 0.2$, trained on OWT) under 128 total denoising steps. Inappropriate tokens are directly corrected without remasking.

cating hyperparameter tuning and hindering practical performance. Moreover, although GIDD enables token revision through additional uniform transitions, its reverse process still retains remasking behavior, which can make correction less direct and reduce the effective correction capacity under few-step parallel generation. To tackle these limitations, we propose a **S**elf-**C**orrecting **D**iscrete **D**iffusion (SCDD) model to reformulate pretrained self-correction with **clear** and **explicit** state transitions in discrete time. Our framework also simplifies the training noise schedule, eliminates a redundant *remasking* step, and relies exclusively on uniform transitions to learn self-correction. To summarize, our contributions are three-fold:

- We redesign the forward process with Signal-to-Noise ratio (SNR) - informed parameters, thus providing separate control over different types of forward noising rates while maintaining clarity in marginal distribution representation at the same time.

- Our training and inference pipeline is clean and engineering-light. More specifically, during training, the model is trained on theoretical ELBO loss without additional re-weighting; during inference, SCDD requires no post-hoc heuristic samplers and no hyperparameter-tuning. All token generation and correction are performed solely by running the backward process derived from Bayes' rule.

- To the best of our knowledge, we are the first to train a diffusion language model that achieves self-correction completely free of *remasking* during generation. Experiments conducted at the GPT-2 scale indicate that our model consistently surpasses existing benchmarks over a few standard datasets, achieving lower generative perplexity without sacrificing sample diversity in parallel generation settings.

**Notation.** Let $\mathbf{m}$ denote the one-hot vector representing the [MASK] token, and let $\mathbf{1}_p$ denote a $p$-dimensional vector of ones. Define the one-hot categorical vocabulary (including the [MASK] token) as

$$\mathcal{V} := \left\{ \mathbf{x} \in \{0,1\}^{K+1} : \sum_{i=1}^{K+1} \mathbf{x}_i = 1_{K+1} \right\},$$

with $|\mathcal{V}| = K+1$. We further define $\mathcal{V}^{/\mathbf{m}} := \mathcal{V} \setminus \{\mathbf{m}\}$ as the vocabulary excluding the [MASK] token. We use $\text{Cat}(\cdot\,; \boldsymbol{\pi})$ to denote the categorical distribution over $\mathcal{V}$ with probability vector $\boldsymbol{\pi} \in \Delta^{K+1}$, where $\Delta^{K+1}$ is the $(K+1)$-simplex.

## 2. Preliminaries

Masked diffusion models (MDM/MDLM) (Sahoo et al., 2024; Shi et al., 2024) are a class of generative models that corrupt data by gradually replacing clean tokens with a special [MASK] token. The forward process starts from a clean token $\mathbf{x} \in \mathcal{V}^{/\mathbf{m}}$ and progressively increases the probability of transitioning to $\mathbf{m}$. Formally, for time points $s, t$ with $0 \leq s < t \leq 1$, the forward kernel is defined as

$$q(\mathbf{z}_t \mid \mathbf{z}_s) = \text{Cat}\left(\mathbf{z}_t; \alpha_{t|s}\,\mathbf{z}_s + (1 - \alpha_{t|s})\,\mathbf{m}\right),$$

where $\alpha_{t|s} = \alpha_t / \alpha_s$ and $\alpha_t \in [0,1]$ is a noise scheduling function controlling the survival rate of the token. The induced marginal is

$$q(\mathbf{z}_t \mid \mathbf{x}) = \text{Cat}\left(\mathbf{z}_t; \alpha_t \mathbf{x} + (1 - \alpha_t)\,\mathbf{m}\right).$$

As $t \to 1$, the process collapses to the prior $q_1(\mathbf{z}_1) = \mathbf{1}_{\{\mathbf{z}_1 = \mathbf{m}\}}$, meaning that the clean token is replaced by the [MASK] token.

Given this forward process, sampling proceeds in the reverse direction using a learned parametric kernel $p_\theta(\mathbf{z}_s \mid \mathbf{z}_t)$ that denoises [MASK] tokens back to clean tokens. The denoiser is trained by minimizing the negative ELBO, which reduces to

$$\mathcal{L} = \mathbb{E}_{t, \mathbf{x}, \mathbf{z}_t} \left[ \frac{\alpha_t'}{1 - \alpha_t} \log\left(\mathbf{x}_\theta(\mathbf{z}_t, t)^\top \mathbf{x}\right) \right],$$

where $\mathbf{x}_\theta(\mathbf{z}_t, t)$ is a denoiser that predicts the clean token distribution.

*Table 1.* Comparison of MDLM, GIDD, and SCDD. SCDD decouples token correction from masking, leading to a cleaner generator and explicit backward dynamics.

| Model | Generator $R_t(\mathbf{z}_t, \mathbf{z}_s), \mathbf{z}_s \neq \mathbf{m}$ | Self-Correction | Remask-Free | Closed-form Backward | Decoupled SNRs |
|---|---|---|---|---|---|
| MDLM | $\dfrac{\gamma_t'}{\gamma_t} \mathbf{z}_t^\top (\mathbf{z}_s - \mathbf{m})$ | ✗ | ✗ | ✓ | – |
| GIDD | $\left(\dfrac{\gamma_t'}{\gamma_t} + \dfrac{\rho_t'}{\rho_t}\right) \mathbf{z}_s^\top \mathbf{z}_t - \mathbf{z}_t^\top \left[\gamma_t \dfrac{\rho_t'}{\rho_t} \mathbf{u} + \left((1 - \gamma_t)\dfrac{\rho_t'}{\rho_t} + \dfrac{\gamma_t'}{\gamma_t}\right) \mathbf{m}\right]$ | ✓ | ✗ | ✗ | ✗ |
| SCDD | $\left(\dfrac{\gamma_t'}{\gamma_t} + \dfrac{\rho_t'}{\rho_t}\right) \mathbf{z}_s^\top \mathbf{z}_t - \mathbf{z}_t^\top \left(\dfrac{\rho_t'}{\rho_t} \mathbf{u} + \dfrac{\gamma_t'}{\gamma_t} \mathbf{m}\right)$ | ✓ | ✓ | ✓ | ✓ |

**Limitations.** Despite its scalable training, MDLM lacks an explicit self-correction mechanism to revise low-quality tokens from earlier steps, degrading the quality of parallel generation (Jiang et al., 2025). Consequently, MDLM typically decodes only a few tokens per step to achieve competitive reasoning performance (Zhao et al., 2025; Rojas et al., 2026), often making it even slower than autoregressive (AR) alternatives with speculative decoding.

## 3. Self-Correcting Discrete Diffusion Model

To maximize parallel generation potential, we reformulate the MDLM pipeline to enhance self-correction during pre-training, a stage known to yield strong generalization performance (Sutskever & Patel, 2025). Motivated by BERT's success in semantic understanding (Devlin et al., 2019), we incorporate uniform transitions alongside the absorbing-mask in the forward process. While related ideas have been explored in prior works, such as GIDD (von Rütte et al., 2025)[1], our approach differs by introducing clear and explicit state transitions and eliminating the need for an additional remasking step completely. This simplification substantially eases hyperparameter tuning and improves parallel generation through enhanced self-correction.

### 3.1. Forward Noising Process

For clarity, we first derive the model for the single token case before generalizing it to the sequence case. We consider a discrete-time forward Markov noising process $q$ that transforms the clean data $\mathbf{x}$ into latent variables $\{\mathbf{z}_{t_i}\}_{i=0}^T$. A typical choice of time grid $\{t_i\}_{i=0}^T$ is equal-spaced grid $\{\frac{i}{T}\}_{i=0}^T$, which is also the one that we use throughout this paper. For notation convenience, we define $\mathbf{z}_{t_{-1}} = \mathbf{x}$ and interchangeably use them to refer to the clean data. The joint distribution of clean data and latent variables follows

$$q(\mathbf{x}, \mathbf{z}_0, ..., \mathbf{z}_1) = q_{\text{data}}(\mathbf{x}) \prod_{i=0}^T q(\mathbf{z}_{t_i} \mid \mathbf{z}_{t_{i-1}}), \quad (1)$$

---

[1] We refer readers to Appendix Section F for further discussion.

where $q_{\text{data}}$ is the unknown data distribution. Let $\mathbf{u} = \frac{1}{K}(\mathbf{1}_{K+1} - \mathbf{m})$ denote the probability vector for the uniform distribution over all non-[MASK] tokens. We use $\mathbf{u}$ to refer to both the vector and the distribution itself. To elicit self-correction capability, we are interested in forward processes whose marginal distributions of $\mathbf{z}_t$ takes the form

$$q(\mathbf{z}_t \mid \mathbf{x}) = \text{Cat}\Big(\mathbf{z}_t; \gamma_t\big(\rho_t \mathbf{x} + (1 - \rho_t)\mathbf{u}\big) + (1 - \gamma_t)\mathbf{m}\Big), \quad (2)$$

where $\rho_t, \gamma_t : [0, 1] \rightarrow [0, 1]$ with boundary conditions $\rho_1 = \gamma_1 = 0$ and $\rho_0, \gamma_0 \approx 1$. Moreover, $\rho_0, \gamma_0 \rightarrow 1^-$ as $T \rightarrow \infty$. The mixture of $\mathbf{x}$, $\mathbf{u}$ and $\mathbf{m}$ in the probability vector ensures the diffusion model encounters both uniform noise and [MASK] tokens during training. At time $t = 0$, the latent variable $\mathbf{z}_0$ is slightly corrupted relative to $\mathbf{x}$; however, this corruption probability vanishes as the number of timesteps $T$ approaches infinity. At time $t = 1$, the clean data is completely corrupted and becomes [MASK]. For intermediate times $t \in (0, 1)$, the token can be replaced by another non-[MASK] token or masked.

We refer to $\rho_t \gamma_t$, the probability mixture ratio of $\mathbf{x}$, as the probability that "$\mathbf{z}_t$ *retains* $\mathbf{x}$", meaning that the event $\{\mathbf{z}_t = \mathbf{x}\}$ occurs due to an explicit stay-in mechanism rather than by coincidence from uniform sampling. In particular, this excludes the case where $\mathbf{z}_t$ is sampled from $\mathbf{u}$ and happens to equal $\mathbf{x}$. Similarly, $\gamma_t(1 - \rho_t)$ is referred to as the probability that "$\mathbf{z}_t$ is sampled from $\mathbf{u}$". The parameters can also be individually interpreted as two kinds of signal-to-noise ratios (SNR):

$$\gamma_t = q(\mathbf{z}_t \neq \mathbf{m} \mid \mathbf{x}), \quad \rho_t = q(\mathbf{z}_t \text{ retains } \mathbf{x} \mid \mathbf{z}_t \neq \mathbf{m}, \mathbf{x}),$$

where $\gamma_t$ measures the SNR of absorbing mask, and $\rho_t$ measures the SNR of uniform transitions. For notation convenience, we set $\rho_{-1} = \gamma_{-1} = 1$.

There are infinite forward noising processes with marginals agree with (2). However, to eliminate the remasking behavior during token generation (i.e., the backward process), the [MASK] state $\mathbf{m}$ needs to be an absorbing state of the forward process. Therefore, we make an additional assumption

that $\rho_t, \gamma_t$ are monotonically decreasing sequences. Under this mild assumption, the marginal in (2) can be realized by a Markov chain of which $\mathbf{m}$ is an absorbing state, similar to masked diffusion models (Sahoo et al., 2024; Shi et al., 2024).

*Proposition 1.* When $\rho_t, \gamma_t$ are monotonically decreasing, the following forward Markov transition kernel induces the marginal distribution (2):

$$q(\mathbf{z}_t \mid \mathbf{z}_s) := \begin{cases} \gamma_{t|s}\left(\mathbf{1}_{\{\mathbf{z}_s=\mathbf{z}_t\}}\rho_{t|s} + \frac{1-\rho_{t|s}}{K}\right), & \text{if } \mathbf{z}_t, \mathbf{z}_s \neq \mathbf{m}, \\ 1 - \gamma_{t|s}, & \text{if } \mathbf{z}_t = \mathbf{m}, \mathbf{z}_s \neq \mathbf{m}, \\ 1, & \text{if } \mathbf{z}_t = \mathbf{z}_s = \mathbf{m}, \end{cases} \tag{3}$$

where $t, s \in \{t_{-1}, t_0, ..., t_T\}$ are two adjacent time points satisfying $t > s$, $\rho_{t|s} := \frac{\rho_t}{\rho_s}$, and $\gamma_{t|s} := \frac{\gamma_t}{\gamma_s}$.

*Proof.* Since $\rho_t$ and $\gamma_t$ are monotonically decreasing, the above transition kernel is well-defined. The Markovian forward process implied by the forward kernel (3) can be equivalently characterized as

$$q(\mathbf{z}_t \mid \mathbf{z}_s) = \text{Cat}\left(\mathbf{z}_t; \mathbf{z}_s^\top Q_{t|s}\right)$$

where

$$Q_{t|s} = \begin{pmatrix} \rho_{t|s}\gamma_{t|s}\mathbf{I} + \frac{(1-\rho_{t|s})\gamma_{t|s}}{K}\mathbf{1}\mathbf{1}^\top & (1-\gamma_{t|s})\mathbf{1} \\ \mathbf{0}^\top & 1 \end{pmatrix} \tag{4}$$

is the forward transition matrix, where $\mathbf{I}$ and $\mathbf{1}$ are abbreviations of the $K$-dimensional identity matrix and $K$-dimensional all-ones vector. Let $i$ be the unique index such that $t_i = t$, the marginal distribution of $\mathbf{z}_t$ is therefore

$$q(\mathbf{z}_t \mid \mathbf{x}) = \text{Cat}(\mathbf{z}_t; \mathbf{x}^\top \bar{Q}_t), \tag{5}$$

where

$$\begin{aligned} \bar{Q}_t &= \prod_{j=0}^{i} Q_{t_j|t_{j-1}} \\ &= \begin{pmatrix} \bar{A}_t & (1 - \prod_{j=0}^{i}\gamma_{t_j|t_{j-1}})\mathbf{1} \\ \mathbf{0}^\top & 1 \end{pmatrix} \\ &= \begin{pmatrix} \bar{A}_t & (1-\gamma_t)\mathbf{1} \\ \mathbf{0}^\top & 1 \end{pmatrix}. \end{aligned}$$

$\bar{A}_t$ is greatly simplified due to the idempotence [2]

$$\begin{aligned} \bar{A}_t &= \prod_{j=0}^{i}\left(\rho_{t_j|t_{j-1}}\gamma_{t_j|t_{j-1}}\mathbf{I} + \frac{(1-\rho_{t_j|t_{j-1}})\gamma_{t_j|t_{j-1}}}{K}\mathbf{1}\mathbf{1}^\top\right) \\ &= \left(\prod_{j=0}^{i}\rho_{t_j|t_{j-1}}\gamma_{t_j|t_{j-1}}\right)\mathbf{I} \\ &\quad + \frac{1}{K}\left(\prod_{j=0}^{i}\gamma_{t_j|t_{j-1}} - \prod_{j=0}^{i}\rho_{t_j|t_{j-1}}\gamma_{t_j|t_{j-1}}\right)\mathbf{1}\mathbf{1}^\top \\ &= \gamma_t\left(\rho_t I + \frac{1}{K}(1-\rho_t)\mathbf{1}\mathbf{1}^\top\right), \end{aligned}$$

Therefore, substituting the closed-form expression of $\bar{Q}_t$ into (5), we obtain that, for any ordinary token $\mathbf{z}_t \neq \mathbf{m}$,

$$q(\mathbf{z}_t \mid \mathbf{x}) = \gamma_t\left(\rho_t\mathbf{1}_{\{\mathbf{z}_t=\mathbf{x}\}} + \frac{1-\rho_t}{K}\right),$$

while

$$q(\mathbf{z}_t = \mathbf{m} \mid \mathbf{x}) = 1 - \gamma_t.$$

This is exactly the marginal distribution specified in (2). $\square$

**Forward Process in Continuous Time.** The discrete-time Markov chain introduced above can be extended to continuous time via a continuous-time Markov chain (CTMC) model. Although deriving SCDD doesn't require CTMC theory, we include this analysis for the sake of completeness. Moreover, when we borrow the forward transition rate concept, we obtain a better interpretation of the noise schedules of our model. As before, we assume that $\gamma_t$ and $\rho_t$ are differentiable, decreasing functions of time $t \in [0,1]$.

Let $s$ and $t$ be two adjacent time points with $\Delta t = t - s > 0$. We define the infinitesimal generator $R_t(\cdot, \cdot)$ by

$$R_t(\mathbf{z}_t, \mathbf{z}_s) := \begin{cases} \left(\frac{\gamma_t'}{\gamma_t} + \frac{\rho_t'}{\rho_t}\right)\mathbf{z}_t^\top\mathbf{z}_s \\ -\mathbf{z}_t^\top\left(\frac{\rho_t'}{\rho_t}\mathbf{u} + \frac{\gamma_t'}{\gamma_t}\mathbf{m}\right), & \mathbf{z}_s \neq \mathbf{m} \\ 0, & \mathbf{z}_s = \mathbf{m}. \end{cases} \tag{6}$$

The following lemma proves that (6) is the forward transition rate of the process (3). We leave the complete proof to Appendix A.

*Lemma 2.* Let $R_t$ be defined as in (6). Then, $R_t$ is a valid infinitesimal generator (rate matrix) in the sense of Lipman et al. (2024); Gat et al. (2024), i.e.,

$$R_t(\mathbf{z}_t, \mathbf{z}_s) \geq 0, \forall \mathbf{z}_t \neq \mathbf{z}_s, \quad \sum_{\mathbf{z}_t} R_t(\mathbf{z}_t, \mathbf{z}_s) = 0.$$

Moreover, the discrete-time forward kernel (3) admits the continuous-time expansion:

$$q(\mathbf{z}_t \mid \mathbf{z}_s) = \delta_{\mathbf{z}_t, \mathbf{z}_s} + \Delta t\, R_t(\mathbf{z}_t, \mathbf{z}_s) + o(\Delta t).$$

---

[2]Define $\mathbf{P} := \mathbf{1}\mathbf{1}^\top/K$. Since $\mathbf{P}^2 = \mathbf{P}$, for any scalars $a, b$, we have $(a\mathbf{I} + (1-a)\mathbf{P})(b\mathbf{I} + (1-b)\mathbf{P}) = ab\mathbf{I} + (1-ab)\mathbf{P}$.

Therefore, the probability evolution induced by $R_t$ is the continuous-time limit of the discrete Markov chain (3).

*Remark* 3. When $\mathbf{z}_s \neq \mathbf{m}$, $\rho_t$ and $\gamma_t$ govern the forward transition rates for sampling $\mathbf{z}_t$ from $\mathbf{u}$ and setting $\mathbf{z}_t$ to [MASK], respectively. Thus, in addition to their SNR interpretation in the marginals, they act as explicit and independent controllers of the two noise mechanisms. In contrast, under the same marginal representation, GIDD couples the absorbing-mask and uniform-transition components in its forward rates, preventing independent control without compromising the clean marginal form. SCDD preserves this marginal structure while decoupling the two transition rates. See Table 1 for a generator-level comparison and Appendices A.1 and D.2 for further discussion.

## 3.2. Backward Denoising Process

Follow the same notations, the true posterior $q(\mathbf{z}_s \mid \mathbf{z}_t, \mathbf{x})$ derived from Bayes' rule is given by:

$$q(\mathbf{z}_s \mid \mathbf{z}_t, \mathbf{x}) =$$
$$\begin{cases} \frac{\rho_s \mathbf{z}_s^\top \mathbf{x} + (1-\rho_s)\frac{1}{K}}{\rho_t \mathbf{z}_t^\top \mathbf{x} + (1-\rho_t)\frac{1}{K}} \left( \frac{\rho_t}{\rho_s} \mathbf{z}_s^\top \mathbf{z}_t + \frac{\rho_s - \rho_t}{\rho_s}\frac{1}{K} \right), & \mathbf{z}_t \neq \mathbf{m} \\ (1 - \mathbf{z}_s^\top \mathbf{m})\frac{\gamma_s - \gamma_t}{1-\gamma_t} \left( \rho_s \mathbf{z}_s^\top \mathbf{x} + (1-\rho_s)\frac{1}{K} \right) \\ \quad + \mathbf{z}_s^\top \mathbf{m}\frac{1-\gamma_s}{1-\gamma_t}, & \mathbf{z}_t = \mathbf{m} \end{cases}$$
$$(7)$$

We refer to the Appendix B for the details of the derivation.

**Efficient Parallel Self-Correction.** Since we have [MASK] being as an absorbing state in the forward process, it follows that there is no transition from non-[MASK] tokens to [MASK], i.e., no remasking, in the backward process. By eliminating the redundant remasking step, SCDD acquires more correction capacity for any given number of inference steps. In particular, SCDD can be twice as efficient as purely remasking-based self-correcting discrete diffusion models, since it only needs one step to correct a token instead of two. As a result, it potentially improves the generation quality, especially in few-step generation scenarios.

Since we don't know the true clean data $\mathbf{x}$, we follow previous work (Sohl-Dickstein et al., 2015; Sahoo et al., 2024) to parameterize the backward process as

$$p_\theta(\mathbf{z}_s \mid \mathbf{z}_t) := q(\mathbf{z}_s \mid \mathbf{z}_t, \mathbf{x} = \mathbf{x}_\theta(\mathbf{z}_t, t))$$
$$= \begin{cases} \frac{\rho_s \mathbf{z}_s^\top \mathbf{x}_\theta + (1-\rho_s)\frac{1}{K}}{\rho_t \mathbf{z}_t^\top \mathbf{x}_\theta + (1-\rho_t)\frac{1}{K}} \left( \frac{\rho_t}{\rho_s} \mathbf{z}_s^\top \mathbf{z}_t + \frac{\rho_s - \rho_t}{\rho_s}\frac{1}{K} \right), & \mathbf{z}_t \neq \mathbf{m}, \\ (1 - \mathbf{z}_s^\top \mathbf{m})\frac{\gamma_s - \gamma_t}{1-\gamma_t} \left( \rho_s \mathbf{z}_s^\top \mathbf{x}_\theta + (1-\rho_s)\frac{1}{K} \right) \\ \quad + \mathbf{z}_s^\top \mathbf{m}\frac{1-\gamma_s}{1-\gamma_t}, & \mathbf{z}_t = \mathbf{m}, \end{cases}$$
$$(8)$$

where

$$\mathbf{x}_\theta(\cdot, \cdot) : \mathcal{V} \times [0,1] \to \Delta^{K+1},$$
$$(\mathbf{z}_t, t) \mapsto \mathbf{x}_\theta(\mathbf{z}_t, t),$$

is a denoising neural network that predicts clean data $\mathbf{x}$ given the latent variable $\mathbf{z}_t$ at time $t$. We omit the arguments of $\mathbf{x}_\theta$ when no confusion arises. Such parameterization indeed defines a valid density $p_\theta(\cdot \mid \mathbf{z}_t)$ conditional $\mathbf{z}_t$, see Appendix B.2 for more details. We also derive the corresponding backward transition rate under CTMC theory in Appendix B.3 for completeness.

We further impose the *Zero Masking Probabilities* constraint on the denoising network $\mathbf{x}_\theta$ as in MDLM (Sahoo et al., 2024), i,e.,

$$\mathbf{x}_\theta(\mathbf{z}_t, t)^\top \mathbf{m} = 0, \quad \forall \, \mathbf{z}_t, t.$$

However, we release the *Carry-Over Unmasking* constraint in MDLM to allow token self-correction during generation: even if $\mathbf{z}_t \neq \mathbf{m}$, we still allow $\mathbf{x}_\theta$ to assign nonzero probability mass to tokens other than $\mathbf{z}_t$ so that previously unmasked tokens can be revised in later denoising steps.

Additionally, we set the prior distribution to be $p_\theta(\mathbf{z}_1) := \mathbf{1}(\mathbf{z}_1 = \mathbf{m})$, indicating that we start the backward process from [MASK]. With these parameterizations, we define the generative model of $\mathbf{x}$ as:

$$p_\theta(\mathbf{x}) := \int p_\theta(\mathbf{z}_1) p_\theta(\mathbf{x} \mid \mathbf{z}_0) \prod_{i=1}^{T} p_\theta(\mathbf{z}_{t_{i-1}} \mid \mathbf{z}_{t_i}) \mathrm{d}\mathbf{z}_{0:1}$$
$$(9)$$

## 3.3. ELBO

To train a diffusion model, we leverage variational inference and minimize the usual negative evidence lower bound (NELBO) as in previous works (Sohl-Dickstein et al., 2015; Ho et al., 2020; Sahoo et al., 2024). The discrete-time NELBO for finite $T$ is given by:

$$\mathcal{L}_{\mathrm{NELBO}}^T := \underbrace{-\mathbb{E}_q\big[ \log p_\theta(\mathbf{x} \mid \mathbf{z}_0) \big]}_{\mathcal{L}_{\mathrm{reconstruction}}^T}$$
$$+ \underbrace{\mathbb{E}_q\big[ D_{\mathrm{KL}}\big( q(\mathbf{z}_1 \mid \mathbf{x}) \| p_\theta(\mathbf{z}_1) \big) \big]}_{\mathcal{L}_{\mathrm{prior}}}$$
$$+ \underbrace{\mathbb{E}_q\left[ \sum_{i=1}^{T} D_{\mathrm{KL}}\big( q(\mathbf{z}_{t_{i-1}} \mid \mathbf{z}_{t_i}, \mathbf{x}) \| p_\theta(\mathbf{z}_{t_{i-1}} \mid \mathbf{z}_{t_i}) \big) \right]}_{\mathcal{L}_{\mathrm{diffusion}}^T},$$
$$(10)$$

where $q$ is the abbreviation of $q(\mathbf{x}, \mathbf{z}_{0:1})$ defined in (1), and $\mathcal{L}_{\mathrm{diffusion}}^T$ is given by (11) up to $\theta$-independent additive constants. A distinctive feature of our loss function is that the latent state $\mathbf{z}_t$ contributes to the gradient regardless of whether it is masked ($\mathbf{z}_t = \mathbf{m}$) or unmasked ($\mathbf{z}_t \neq \mathbf{m}$), which is similar to GIDD (von Rütte et al., 2025). Note that if $\rho_t \equiv 1$, the diffusion loss (11) is reduced to MDLM

$$\mathcal{L}_{\text{diffusion}}^T = \begin{cases} -\mathbb{E}_{t\sim\mathcal{U}\{t_1,\dots,t_T\}}\mathbb{E}_q\left[T\sum_{\mathbf{v}\neq\mathbf{m}}\frac{\rho_s\mathbf{v}^\top\mathbf{x}+(1-\rho_s)\frac{1}{K}}{\rho_t\mathbf{z}_t^\top\mathbf{x}+(1-\rho_t)\frac{1}{K}}\left(\frac{\rho_t}{\rho_s}\mathbf{v}^\top\mathbf{z}_t+\frac{\rho_s-\rho_t}{\rho_s}\frac{1}{K}\right)\log\frac{\rho_s\mathbf{v}^\top\mathbf{x}_\theta+(1-\rho_s)\frac{1}{K}}{\rho_t\mathbf{z}_t^\top\mathbf{x}_\theta+(1-\rho_t)\frac{1}{K}}\right], & \text{if } \mathbf{z}_t\neq\mathbf{m}, \\ -\mathbb{E}_{t\sim\mathcal{U}\{t_1,\dots,t_T\}}\mathbb{E}_q\left[T\sum_{\mathbf{v}\neq\mathbf{m}}\frac{\gamma_s-\gamma_t}{1-\gamma_t}(\rho_s\mathbf{v}^\top\mathbf{x}+(1-\rho_s)\frac{1}{K})\log\left(\rho_s\mathbf{v}^\top\mathbf{x}_\theta+(1-\rho_s)\frac{1}{K}\right)\right], & \text{if } \mathbf{z}_t=\mathbf{m}. \end{cases} \tag{11}$$

$$\mathcal{L}_{\text{diffusion}}^\infty = \begin{cases} \mathbb{E}_{t\sim\mathcal{U}[0,1]}\mathbb{E}_q\left[\sum_{\mathbf{v}\neq\mathbf{z}_t,\mathbf{m}}\left(\frac{(\rho_t\mathbf{v}^\top\mathbf{x}+(1-\rho_t)\frac{1}{K})\frac{\rho_t'}{\rho_t}\frac{1}{K}}{\rho_t\mathbf{z}_t^\top\mathbf{x}+(1-\rho_t)\frac{1}{K}}\log\frac{\rho_t\mathbf{v}^\top\mathbf{x}_\theta+(1-\rho_t)\frac{1}{K}}{\rho_t\mathbf{z}_t^\top\mathbf{x}_\theta+(1-\rho_t)\frac{1}{K}}\right)-\frac{\rho_t'(-\mathbf{z}_t^\top\mathbf{x}_\theta+\frac{1}{K})}{\rho_t\mathbf{z}_t^\top\mathbf{x}_\theta+(1-\rho_t)\frac{1}{K}}\right], & \text{if } \mathbf{z}_t\neq\mathbf{m}, \\ \mathbb{E}_{t\sim\mathcal{U}[0,1]}\mathbb{E}_q\left[\sum_{\mathbf{v}\neq\mathbf{m}}\frac{\gamma_t'}{1-\gamma_t}(\rho_t\mathbf{v}^\top\mathbf{x}+(1-\rho_t)\frac{1}{K})\log(\rho_t\mathbf{v}^\top\mathbf{x}_\theta+(1-\rho_t)\frac{1}{K})\right], & \text{if } \mathbf{z}_t=\mathbf{m}. \end{cases} \tag{12}$$

diffusion loss (Sahoo et al., 2024) since the forward noising process doesn't involve uniform noise anymore. We left the derivation of discrete-time ELBO to Appendix C, and the discussion of the relationship with MDLM to Appendix D.1.

**Continuous-time ELBO.** By taking $T\to\infty$, the discrete-time negative ELBO will converge to its continuous-time extension, denoted by $\mathcal{L}_{\text{NELBO}}^\infty$. This will give tight approximation to ELBO (Kingma et al., 2021; Sahoo et al., 2024). Among the three components in (10), $\mathcal{L}_{\text{prior}}\equiv 0$ by construction. When $T\to\infty$, $\lim_{T\to\infty}\mathcal{L}_{\text{reconstruction}}^T = 0$ due to the fact that $\mathbf{z}_0$ converges in distribution to $\mathbf{x}$ when $T\to\infty$. Therefore, the only component left in $\mathcal{L}_{\text{NELBO}}^\infty$ is $\mathcal{L}_{\text{diffusion}}^\infty := \lim_{T\to\infty}\mathcal{L}_{\text{diffusion}}^T$, which is given in (12) up to $\theta$-independent additive constants. See Appendix C for derivation of continuous-time ELBO.

### 3.4. Generalization to Sequences of Tokens

In practice, we operate on sequences of tokens rather than single tokens. Our generalization follows Austin et al. (2021); Sahoo et al. (2024); von Rütte et al. (2025). Let $\mathbf{x}^{1:L}$ be a length-$L$ sequence of clean data, and $\mathbf{z}_t^{1:L}$ be a length-$L$ sequence of latent variables at time $t$. The forward noising process is applied to each of the $L$ positions in the sequence independently to corrupt the clean data $\mathbf{x}^{1:L}$. The model $\mathbf{x}_\theta$ now accepts a whole sequence $\mathbf{z}_t^{1:L}$ as input, and outputs $\mathbf{x}_\theta(\mathbf{z}_t^{1:L}, t) = (\mathbf{x}_\theta^l(\mathbf{z}_t^{1:L}, t))_{l=1}^L$, where $\mathbf{x}_\theta^l(\mathbf{z}_t^{1:L}, t)$ is the predicted distribution of clean data at position $l$. The denoising process admits a factorization over all positions, i.e., $p_\theta(\mathbf{z}_s^{1:L} \mid \mathbf{z}_t^{1:L}) = \prod_{l=1}^L p_\theta\left(\mathbf{z}_s^l \mid \mathbf{z}_t^{1:L}\right)$. The final training objective is defined as the sum of the per-token NELBOs across all positions.

### 3.5. Sampling

We perform the usual ancestral sampling as in other works in this field (Shi et al., 2024; Sahoo et al., 2024; von Rütte et al., 2025; Wang et al., 2026). To sample from the model

$p_\theta(\mathbf{x})$, the backward process is initiated from a sequence with [MASK] at all positions, and runs iteratively according to $p_\theta(\mathbf{z}_s^{1:L} \mid \mathbf{z}_t^{1:L})$ following the prefixed sampling time schedule. Given the current intermediate sample $\mathbf{z}_t^{1:L}$, the model samples in parallel across all positions in the sequence for the next time step $s < t$ in the schedule.

## 4. Experiments

### 4.1. Experiment Setup

We focus on applying SCDD to language modeling tasks, and select two standard datasets, LM1B (One Billion Words Dataset, Chelba et al. (2013)) and the OWT (OpenWebText, Gokaslan & Cohen (2019)), to evaluate SCDD. We use GPT-2 tokenizer, and adopt a DiT (Peebles & Xie, 2023) backbone at SMALL scale throughout the experiments. The model is trained on 33B tokens for LM1B, and 131B tokens for OWT.

To ensure fair comparison, we use the noise schedule that induces the same marginal distribution as GIDD during SCDD training. As for baselines, we retrain GIDD (von Rütte et al., 2025) with $p_u \in \{0.1, 0.2\}$ and MDLM (Sahoo et al., 2024) under similar settings, where $p_u$ is the maximum uniform transition noise ratio (see Appendix E.4). We also implemented ReMDM-cap and ReMDM-conf (Wang et al., 2026) to retrained MDLM as two of the baselines. See Appendix E.1 for training details.

### 4.2. Likelihood Evaluation

Table 2 shows the validation perplexity of each model trained on LM1B and OWT dataset. Similar to the findings in von Rütte et al. (2025), we see the models with uniform noise added during training exhibit a degradation in validation perplexity due to the increased difficulty of learning transitions between non-[MASK] tokens. However, by comparing the best models of SCDD and GIDD, we still see a 3.7% and 9.9% decrease in validation perplexity on

*Table 2.* Validation perplexity (Val PPL) on LM1B and OWT. [†]Reported in Sahoo et al. (2024). [*]Total number of tokens seen, calculated as double of the average number of tokens seen as reported in Sahoo et al. (2024).

| MODEL (SMALL) | VAL PPL. ($\downarrow$) | |
| --- | --- | --- |
| | LM1B | OWT |
| MDLM[†]($66B^*$) | 27.04 | - |
| MDLM[†]($524B^*$) | - | 23.21 |
| MDLM | 32.98 | 24.72 |
| GIDD+ ($p_u = 0.1$) | 39.98 | 31.54 |
| GIDD+ ($p_u = 0.2$) | 40.70 | 32.19 |
| SCDD ($p_u = 0.1$, OURS) | 39.16 | 28.41 |
| SCDD ($p_u = 0.2$, OURS) | 46.54 | 32.49 |

LM1B and OWT, respectively. Indeed, SCDD is trained without having to learn the transitions from non-[MASK] tokens to the [MASK] token in the backward process, which slightly eases the training task. We will see later that introducing uniform noise substantially enhances the model's self-correction capability with this modest increase in validation perplexity.

### 4.3. Unconditional Language Generation

To evaluate the model's ability to learn from large-scale language data and to assess the effectiveness of self-correction at inference time, we perform unconditional text generation and report the generative perplexity (Gen PPL), a common quality metric for text generation, evaluated by the GPT2-large model in Table 3. As noted in Zheng et al. (2025), Gen PPL can be extremely low if the model produces highly repetitive and redundant tokens. Therefore, we also report the unigram entropy metric in Appendix E.2 in addition to Gen PPL, serving as a sanity check for the diversity of generated texts.

SCDD ($p_u = 0.2$) consistently outperforms the best GIDD+ baseline across all denoising steps on LM1B and OWT with a comparable entropy. Notably, SCDD has significant improvements over baselines at few-step parallel generation scenarios, with 55% and 9.2% decrease in Gen PPL when compared to ReMDM-cap and GIDD+ at 32-step generation, respectively. We attribute the performance gain to two primary factors. First, GIDD+ utilizes a dynamic weighting scheme to mitigate training loss explosion. However, this scheme downweights the training samples at boundaries ($t \to 0$ and $t \to 1$), which may impede the model from learning critical final corrections during generation. Second, GIDD+ does not eliminate remasking during inference, which likely reduces the capacity of self-correction compared to SCDD.

**LLM-as-a-judge Evaluation.** To provide stronger semantic evidence showing that the SCDD outputs are actually

better, we perform *LLM-as-a-judge* evaluation to directly assess correction quality beyond Gen PPL and entropy. For each setting, we corrupt 256 clean OWT sequences from $t = 0$ to $t = 0.8$ using SCDD's forward process, then let both models denoise from the identical corrupted input. Each model generates outputs at 6 different total step counts with nucleus sampling ($p = 0.9$). The GPT-5.4 judge scores each pair on five dimensions (*Clarity*, *Grammaticality*, *Factuality*, *Style*, *Creativity*) on a 1 to 10 scale. Finally, the model decides the "winning text".

We report per-metric scores with paired $t$-tests and overall win rates with binomial tests ($n = 256$). The evaluation prompt for GPT-5.4 is in Figure 4, strictly following von Rütte et al. (2025). Results are presented in Table 4, showing that SCDD consistently outperforms GIDD+ across *Clarity*, *Factuality*, and *Style* metrics, with significant gains in *Clarity* and *Style* at higher step counts. While GIDD+ exhibits a persistent advantage in Creativity, SCDD's ability to maintain higher overall win rates—reaching 60.6% at 1024 steps—highlights its self-correction efficacy in matched-noise environments. In addition to this matched-schedule setting, we see similar results in a cross-ratio setting, where we compare the best-performing SCDD ($p_u = 0.2$) against best GIDD+ ($p_u = 0.1$). See Appendix E.2 and Table 9 for more details.

**Correction Capacity.** To compare the correction capacity between GIDD+ and SCDD, we use the models trained on OWT data, and generate 128 samples for each model under different denoising steps, then calculate the average of *Correction Rate* defined as follows:

$$Correction\ Rate = \frac{C}{L},$$

where $C = $ # Total Corrections, and $L = $ # Context Length. Indeed, from Table 5 we see that SCDD not only achieves significantly higher *Correction Rate*, but also scales faster to a *Correction Rate* of 0.75 at 1024 steps, thus leveraging additional denoising steps to refine the generated texts more efficiently.

### 4.4. Ablation Study

We carry out three ablation studies to investigate into self-correction behaviors. Specifically, we want to know: 1) If a larger uniform noise ratio would encourage more aggressive (parallel) self-correction? 2) How does the timing of peak uniform noise affect the temporal strength of self-correction throughout the generation process? 3) Does "higher correction rate" imply more successful self-corrections, or only more frequent but pointless token revisions?

**Parallel Self-Correction.** To answer the first question, we train our model on Wikitext-103 (Merity et al., 2017) with

*Table 3.* Generative perplexity (Gen PPL) on LM1B and OWT datasets across sampling steps. Lower is better. Bold values indicate the best performance per column. [†]We train all models on OWT with a context length of 512 to be consistent with von Rütte et al. (2025), different from the 1024 context length in Sahoo et al. (2024); Wang et al. (2026).

| GEN. PPL ($\downarrow$) | LM1B (STEPS) | | | | | OWT (STEPS) | | | | | |
|---|---|---|---|---|---|---|---|---|---|---|---|
| MODEL | 16 | 32 | 64 | 128 | 256 | 32 | 64 | 128 | 256 | 512 | 1024 |
| MDLM[†] | 226.0 | 162.6 | 136.7 | 123.0 | 118.6 | 169.9 | 123.6 | 104.7 | 94.8 | 91.9 | 88.5 |
| REMDM-CAP 0.01 | 222.1 | 157.5 | 127.0 | 108.9 | **96.8** | 166.3 | 120.9 | 95.9 | 81.7 | 73.9 | 68.3 |
| REMDM-CONFIDENCE | 221.1 | 159.5 | 129.8 | 122.8 | 120.4 | 167.6 | 118.3 | 98.1 | 87.9 | 83.9 | 80.5 |
| GIDD+ ($p_u = 0.1$) | 171.1 | 146.4 | 134.9 | 131.9 | 128.7 | 82.1 | 71.4 | 66.7 | 65.0 | 64.8 | 63.8 |
| GIDD+ ($p_u = 0.2$) | 192.7 | 165.5 | 151.9 | 147.3 | 144.8 | 90.5 | 79.0 | 75.1 | 73.2 | 72.0 | 71.2 |
| SCDD ($p_u = 0.1$, OURS) | 159.8 | 133.5 | 119.2 | 113.7 | 108.9 | 78.6 | 71.8 | 67.6 | 66.0 | 63.6 | 61.3 |
| SCDD ($p_u = 0.2$, OURS) | **159.2** | **130.0** | **115.2** | **108.4** | 102.6 | **74.5** | **67.1** | **60.7** | **59.6** | **58.2** | **55.7** |

*Table 4.* Matched-schedule Setting — SCDD ($p_u$=0.2) vs GIDD+ ($p_u$=0.2). Values are formatted as SCDD (GIDD+). Significance: [*]$p < 0.05$, [**]$p < 0.01$.

| METRICS | STEPS | | | | | |
|---|---|---|---|---|---|---|
| | 32 | 64 | 128 | 256 | 512 | 1024 |
| CLARITY | 1.65 (1.49)** | 1.62 (1.52) | 1.69 (1.50)** | 1.68 (1.56)* | 1.73 (1.50)** | 1.73 (1.48)** |
| GRAMM. | 1.38 (1.42) | 1.46 (1.49) | 1.45 (1.46) | 1.49 (1.53) | 1.51 (1.47) | 1.57 (1.46)* |
| FACT. | 2.20 (2.11)* | 2.13 (2.07) | 2.20 (2.05)** | 2.21 (2.12)* | 2.13 (1.97)** | 2.20 (2.07)** |
| STYLE | 1.62 (1.50)* | 1.63 (1.53) | 1.66 (1.51)** | 1.66 (1.57) | 1.70 (1.52)** | 1.73 (1.50)** |
| CREATIVITY | 2.78 (2.99)** | 2.81 (3.02)** | 2.88 (3.14)** | 2.84 (3.14)** | 2.94 (3.14)** | 2.93 (3.15)** |
| WIN RATE | 55.9% | 53.0% | 55.3% | 52.0% | 58.1%* | 60.6%** |

*Table 5.* Comparison of *Correction Rate* between GIDD and SCDD across denoising steps ($N$).

| MODEL ($p_u = 0.2$) | TOTAL # DENOISING STEPS $N$ ($\uparrow$) | | | | |
|---|---|---|---|---|---|
| | 64 | 128 | 256 | 512 | 1024 |
| GIDD+ | 0.39 | 0.40 | 0.40 | 0.40 | 0.40 |
| SCDD (OURS) | 0.69 | 0.71 | 0.72 | 0.73 | 0.75 |

maximum uniform noise ratio $p_u \in \{1e\text{-}6, 0.05, 0.1, 0.2\}$, see Appendix E.1 for training details. Then, we define the following metric:

$$Correction\ Rate\ per\ Step = \frac{C}{L \times N},$$

where $C$ = # Total Corrections, $L$ = # Context Length, and $N$ = # Denoising Steps. Note that the *Correction Rate per Step* is derived by applying a normalization factor $N$ to the standard *Correction Rate*. This normalization compensates for the inherent bias toward higher total corrections in generation with more denoising steps as we see in Table 5.

For a fixed $N$, Figure 2 shows that total corrections increases as $p_u$ increases, consistent with the intuition that more uniform noise in the forward process would incur more corrections in the backward process. Notably, *Correction Rate per*

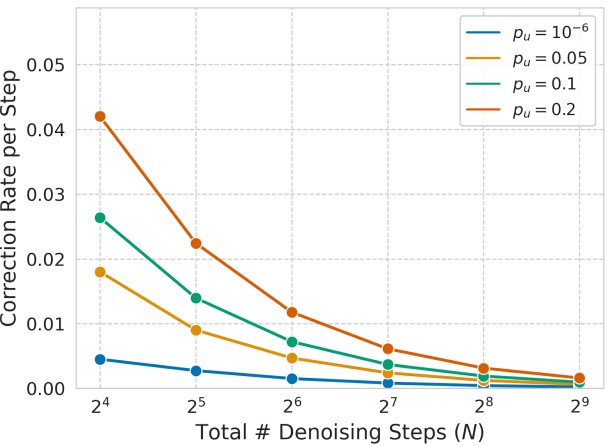

*Figure 2. Correction Rate per Step* versus total number of denoising steps at different maximum uniform noise ratios. Reported values are averaged from 128 independently generated sequences.

*Step* is negatively correlated with the number of total denoising steps, indicating more active parallel self-correction at fewer-step generation scenarios. In contrast, self-correction is amortized throughout the whole process in an extended sampling horizon, resulting in a modest increase in total corrections.

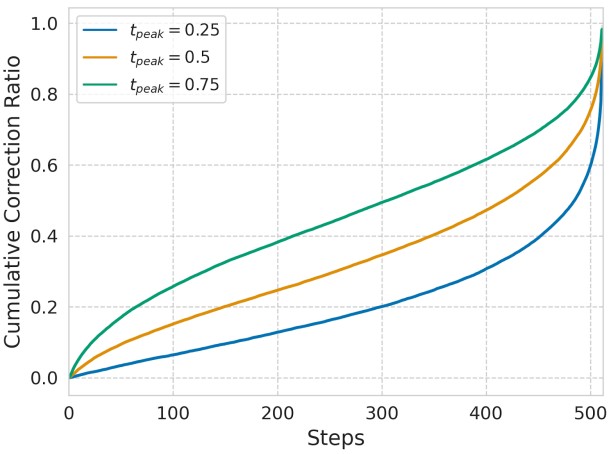

*Figure 3.* Cumulative correction ratio over 512 denoising steps, defined as the fraction of total corrections completed by step $s$. Results are averaged over 128 independently generated sequences.

**Dynamics of Self-Correction.** We train the same model on Wikitext-103 data, but with a different noise schedule that attains the maximum uniform noise ratio at a general time point $t_{\text{peak}}$ (see Appendix E.4 for a detailed discussion of noise schedules). From Figure 3 We see a clear pattern that the model shifts the timing of its self-correction to align with the peak uniform noise time of the training schedule. When the maximum noise ratio occurs later in the forward process (higher $t_{\text{peak}}$), the model tends to self-correct in the early stage during generation, as shown by the concave trajectory of the $t_{\text{peak}} = 0.75$ curve in the first half. In this setting, the model completes nearly 40% of its total correction within the first 200 steps. Conversely, when the noise peak is shifted to the beginning of the forward process ($t_{\text{peak}} = 0.25$), the correction is significantly delayed, resulting in a convex curve where the majority of the corrections occur only in the final 100 denoising steps.

**Exact Recovery of Corrupted Tokens.** So far, we have seen that SCDD substantially increases the overall correction rate during generation. To directly verify that these corrections are beneficial rather than spurious revisions, we conduct a controlled corruption–recovery experiment on clean OWT validation sequences. For each sequence, we randomly select $K \in \{5, 10, 20, 50\}$ positions, replace their tokens with uniformly sampled alternatives, and apply one SCDD ($p_u = 0.2$) denoising step at the last-step noise level, corresponding to step 127 in a 128-step sampling schedule. We report the *touch rate*, the fraction of corrupted tokens modified by the model, and the *recovery rate*, the fraction of corrupted tokens exactly restored to their original clean values. As shown in Table 6, SCDD touches nearly all corrupted tokens and exactly recovers $64.4\%$–$69.4\%$ of them. Moreover, under the strongest corruption level $K = 50$, Gen PPL drops from 154.8 to 25.5 after one denoising step.

*Table 6.* Recovery rate of corrupted tokens on OWT validation sequences after a single denoising step. Results are averaged over 128 samples with standard errors.

| $K$ | Touch Rate | Recovery Rate | Gen PPL | |
|---|---|---|---|---|
| | | | Corrupted | Corrected |
| 5 | 1.000±0.000 | 0.694±0.013 | 22.0±0.3 | 23.8±0.5 |
| 10 | 1.000±0.000 | 0.652±0.011 | 28.2±0.4 | 24.0±0.4 |
| 20 | 0.999±0.001 | 0.647±0.008 | 44.7±0.7 | 24.3±0.5 |
| 50 | 1.000±0.000 | 0.644±0.005 | 154.8±2.2 | 25.5±0.4 |

These results show that SCDD performs meaningful content recovery rather than merely making frequent token edits.

### 4.5. Benchmark Performance

Finally, we evaluate the models on seven standard common sense benchmarks (ARC-E/C (Clark et al., 2018), BoolQ (Clark et al., 2019), HellaSwag (Zellers et al., 2019), PIQA (Bisk et al., 2020), OBQA (Mihaylov et al., 2018), Winogrande (Sakaguchi et al., 2021)) using the EleutherAI LM Evaluation Harness (Gao et al., 2024) with a batch size of 32, see Appendix E.3 for details. Not surprisingly, we see the models trained on uniform and mask noises underperform the mask-only models on all tasks except ARC-c and OBQA. However, it is important to note that these standard benchmarks primarily measure zero-shot likelihood and do not explicitly reflect the self-correction ability observed in our earlier studies. See Appendix E.3 for a detailed discussion of the results.

## 5. Conclusion

In this work, we propose the Self-Correcting Discrete Diffusion (SCDD) model that enhances self-correction to enable more effective parallel generation. In contrast to post-hoc self-corrective samplers, our approach explicitly learns self-correction during pretraining, leading to improved generalization. Compared to GIDD, our forward transition defines clear and explicit state transitions and removes a redundant remasking step, making the model easier to tune and more effective in few-step parallel generation scenarios. Empirically, SCDD demonstrates better generation performance and stronger parallel self-correction capability on LM1B and OWT. For future work, we are interested in scaling up SCDD to acquire more generalizable self-correction ability, as well as exploring reinforcement learning methods to further enhance self-correction ability in reasoning tasks.

## Impact Statement

This paper presents work whose goal is to advance the field of Machine Learning. There are many potential societal

consequences of our work, none which we feel must be specifically highlighted here.

## Acknowledgements

Guang Lin would like to thank the support of National Science Foundation (DMS-2533878, DMS-2053746, DMS-2134209, ECCS-2328241, CBET-2347401 and OAC-2311848), and U.S. Department of Energy (DOE) Office of Science Advanced Scientific Computing Research program DE-SC0023161, the SciDAC LEADS Institute, and DOE–Fusion Energy Science, under grant number: DE-SC0024583.

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

# A. Proof of Lemma 2

*Proof.* Firstly, $R_t(\mathbf{z}_t, \mathbf{z}_s) \geq 0$ for $\mathbf{z}_t \neq \mathbf{z}_s$ and the row sum condition follows by construction, proving that $R_t$ is a valid infinitesimal generator.

Fix $\mathbf{z}_s \neq \mathbf{m}$ and consider two adjacent time points $s$ and $t = s + \Delta t$. From the discrete forward kernel (3), we have the following three transition probabilities:

$$q(\mathbf{z}_t = \mathbf{m} \mid \mathbf{z}_s) = 1 - \frac{\gamma_t}{\gamma_s},$$

$$q(\mathbf{z}_t \text{ is sampled from } \mathbf{u} \mid \mathbf{z}_s) = \frac{\gamma_t}{\gamma_s}\left(1 - \frac{\rho_t}{\rho_s}\right),$$

$$q(\mathbf{z}_t \text{ retains } \mathbf{z}_s \mid \mathbf{z}_s) = \frac{\gamma_t \rho_t}{\gamma_s \rho_s}.$$

Using first-order Taylor expansions (at $t$) for $\gamma$ and $\rho$,

$$\frac{\gamma_t}{\gamma_s} = 1 + \Delta t \frac{\gamma'_t}{\gamma_t} + o(\Delta t),$$

$$\frac{\rho_t}{\rho_s} = 1 + \Delta t \frac{\rho'_t}{\rho_t} + o(\Delta t).$$

Substituting these gives

$$q(\mathbf{z}_t = \mathbf{m} \mid \mathbf{z}_s) = \Delta t\left(-\frac{\gamma'_t}{\gamma_t}\right) + o(\Delta t).$$

$$q(\mathbf{z}_t \text{ is sampled from } \mathbf{u} \mid \mathbf{z}_s) = \Delta t\left(-\frac{\rho'_t}{\rho_t}\right) + o(\Delta t),$$

$$q(\mathbf{z}_t \text{ retains } \mathbf{z}_s \mid \mathbf{z}_s) = 1 + \Delta t\left(\frac{\gamma'_t}{\gamma_t} + \frac{\rho'_t}{\rho_t}\right) + o(\Delta t),$$

These match exactly the off-diagonal entries of $R_t$ and the diagonal term. Hence

$$q(\mathbf{z}_t \mid \mathbf{z}_s) = \delta_{\mathbf{z}_t, \mathbf{z}_s} + \Delta t\, R_t(\mathbf{z}_t, \mathbf{z}_s) + o(\Delta t),$$

where

$$R_t(\mathbf{z}_t, \mathbf{z}_s) := \begin{cases} \left(\frac{\gamma'_t}{\gamma_t} + \frac{\rho'_t}{\rho_t}\right)\mathbf{z}_t^\top \mathbf{z}_s - \mathbf{z}_t^\top\left(\frac{\rho'_t}{\rho_t}\mathbf{u} + \frac{\gamma'_t}{\gamma_t}\mathbf{m}\right), & \mathbf{z}_s \neq \mathbf{m} \\ 0, & \mathbf{z}_s = \mathbf{m}. \end{cases}$$

$\square$

## A.1. Forward Rate Matrix

Equivalently, the above forward rate admits a compact matrix form. Following mask diffusion models (Sahoo et al., 2024; Shi et al., 2024), we adopt the convention that columns represent current states, and all columns sum to zero. Under this convention, the (time-inhomogeneous) generator matrix can be written as

$$\mathbf{R}_t = \frac{\gamma'_t}{\gamma_t} \underbrace{\left(\mathbf{I} - \mathbf{m}\mathbf{1}^\top\right)}_{\text{MDLM masking structure}} + \frac{\rho'_t}{\rho_t} \underbrace{\left(\mathbf{I} - \mathbf{u}\mathbf{1}^\top\right)\left(\mathbf{I} - \mathbf{m}\mathbf{m}^\top\right)}_{\text{uniform mixing among non-mask tokens}}. \tag{13}$$

The first term is the MDLM masking generator; the second is the standard uniform generator $\left(\mathbf{I} - \mathbf{u}\mathbf{1}^\top\right)$ right-multiplied by $\left(\mathbf{I} - \mathbf{m}\mathbf{m}^\top\right)$, which zeros out the [MASK] column so that [MASK] stays absorbing while uniform transitions only shuffle the $K$ non-mask tokens. Both terms have zero column sums, hence so does $\mathbf{R}_t$.

*Proposition* 4 (Consistency of the matrix generator). Let $\mathbf{R}_t$ be defined in (13). Then the forward rate induced by $\mathbf{R}_t$ via $R_t(\mathbf{z}_t, \mathbf{z}_s) = \mathbf{z}_t^\top \mathbf{R}_t \mathbf{z}_s$ coincides with the forward rate (6).

*Proof.* Using the identity $\mathbf{I} - \mathbf{m}\mathbf{1}^\top = \left(\mathbf{I} - \mathbf{m}\mathbf{1}^\top\right)\left(\mathbf{I} - \mathbf{m}\mathbf{m}^\top\right)$, we can equivalently write (13) as

$$\mathbf{R}_t = \left(\left(\frac{\gamma'_t}{\gamma_t} + \frac{\rho'_t}{\rho_t}\right)\mathbf{I} - \frac{\rho'_t}{\rho_t}\mathbf{u}\mathbf{1}^\top - \frac{\gamma'_t}{\gamma_t}\mathbf{m}\mathbf{1}^\top\right)\left(\mathbf{I} - \mathbf{m}\mathbf{m}^\top\right).$$

If $\mathbf{z}_s = \mathbf{m}$, the trailing factor gives $\left(\mathbf{I} - \mathbf{m}\mathbf{m}^\top\right)\mathbf{m} = \mathbf{0}$, hence $\mathbf{R}_t\mathbf{m} = \mathbf{0}$ and $R_t(\mathbf{z}_t, \mathbf{m}) = 0$ for all $\mathbf{z}_t$, i.e., [MASK] is absorbing.

If $\mathbf{z}_s \neq \mathbf{m}$, then $\mathbf{m}^\top\mathbf{z}_s = 0$ and $\mathbf{1}^\top\mathbf{z}_s = 1$, it follows that $\left(\mathbf{I} - \mathbf{m}\mathbf{m}^\top\right)\mathbf{z}_s = \mathbf{z}_s$, $\left(\mathbf{I} - \mathbf{m}\mathbf{1}^\top\right)\mathbf{z}_s = \mathbf{z}_s - \mathbf{m}$, and $\left(\mathbf{I} - \mathbf{u}\mathbf{1}^\top\right)\mathbf{z}_s = \mathbf{z}_s - \mathbf{u}$. Hence

$$\mathbf{R}_t\mathbf{z}_s = \frac{\gamma'_t}{\gamma_t}\left(\mathbf{z}_s - \mathbf{m}\right) + \frac{\rho'_t}{\rho_t}\left(\mathbf{z}_s - \mathbf{u}\right) = \left(\frac{\gamma'_t}{\gamma_t} + \frac{\rho'_t}{\rho_t}\right)\mathbf{z}_s - \left(\frac{\gamma'_t}{\gamma_t}\mathbf{m} + \frac{\rho'_t}{\rho_t}\mathbf{u}\right),$$

and therefore

$$R_t(\mathbf{z}_t, \mathbf{z}_s) = \mathbf{z}_t^\top\mathbf{R}_t\mathbf{z}_s = \left(\frac{\gamma'_t}{\gamma_t} + \frac{\rho'_t}{\rho_t}\right)\mathbf{z}_t^\top\mathbf{z}_s - \mathbf{z}_t^\top\left(\frac{\rho'_t}{\rho_t}\mathbf{u} + \frac{\gamma'_t}{\gamma_t}\mathbf{m}\right),$$

which matches the scalar expression. $\qquad\square$

*Remark* 5 (degeneration to MDLM). We recall that the forward rate matrix in MDLM (Sahoo et al., 2024) can be written, under the same column-sum-zero convention, as

$$\mathbf{R}_t^{\mathrm{MDLM}} = \frac{(\alpha_t^{\mathrm{MDLM}})'}{\alpha_t^{\mathrm{MDLM}}}\left(\mathbf{I} - \mathbf{m}\mathbf{1}^\top\right) \tag{14}$$

When $\rho_t \equiv 1$ (thus $\rho'_t/\rho_t \equiv 0$), the uniform-mixing term in (13) vanishes and the generator reduces to $\mathbf{R}_t = \frac{\gamma'_t}{\gamma_t}\left(\mathbf{I} - \mathbf{m}\mathbf{1}^\top\right)$, i.e., $\mathbf{R}_t^{\mathrm{MDLM}}$ under the reparameterization $\alpha_t^{\mathrm{MDLM}} = \gamma_t$; see Appendix D.1 for a further discussion of the relation between SCDD and MDLM.

## B. Discussion of Posterior Distribution and Backward Process

In this section, we first derive the posterior distribution in (7). Then, we discuss validity of the backward process defined in (8). Finally, we derive the CTMC backward transition rate.

### B.1. Derivation of Posterior Distribution

Let $s, t$ be two adjacent time points such that $s < t$.

**Case 1: $\mathbf{z}_t \neq \mathbf{m}$.** When $\mathbf{z}_t \neq \mathbf{m}$, we have $\mathbf{z}_s \neq \mathbf{m}$ since SCDD doesn't allow "remasking" during token generation. The posterior $q(\mathbf{z}_s|\mathbf{z}_t, \mathbf{x})$ is given by Bayes' rule:

$$
\begin{aligned}
q(\mathbf{z}_s \mid \mathbf{z}_t, \mathbf{x}) &= \frac{q(\mathbf{z}_s|\mathbf{x})}{q(\mathbf{z}_t|\mathbf{x})}q(\mathbf{z}_t|\mathbf{z}_s) \\
&= \frac{\gamma_s\left(\rho_s\mathbf{1}(\mathbf{z}_s = \mathbf{x}) + (1-\rho_s)\frac{1}{K}\right) + (1-\gamma_s)\mathbf{1}(\mathbf{z}_s = \mathbf{m})}{\gamma_t\left(\rho_t\mathbf{1}(\mathbf{z}_t = \mathbf{x}) + (1-\rho_t)\frac{1}{K}\right) + (1-\gamma_t)\mathbf{1}(\mathbf{z}_s = \mathbf{m})} \cdot \frac{\gamma_t}{\gamma_s}\left[\frac{\rho_t}{\rho_s}\mathbf{1}(\mathbf{z}_s = \mathbf{z}_t) + \frac{\rho_s - \rho_t}{\rho_s}\frac{1}{K}\right]\mathbf{1}(\mathbf{z}_s \neq \mathbf{m}) \\
&= (1 - \mathbf{1}(\mathbf{z}_s = \mathbf{m}))\frac{\rho_s\mathbf{1}(\mathbf{z}_s = \mathbf{x}) + (1-\rho_s)\frac{1}{K}}{\rho_t\mathbf{1}(\mathbf{z}_t = \mathbf{x}) + (1-\rho_t)\frac{1}{K}}\left[\frac{\rho_t}{\rho_s}\mathbf{1}(\mathbf{z}_s = \mathbf{z}_t) + \frac{\rho_s - \rho_t}{\rho_s}\frac{1}{K}\right]. \tag{15}
\end{aligned}
$$

Note that when $\rho_s = 0$, we have $\rho_t = 0$ since $\rho_t$ is nonnegative and nonincreasing. In this case, we use convention $\frac{\rho_t}{\rho_s} = 0$ and the posterior becomes $q(\mathbf{z}_s|\mathbf{z}_t, \mathbf{x}) \equiv \frac{1}{K}$.

**Case 2: $\mathbf{z}_t = \mathbf{m}$.** If $\mathbf{z}_s = \mathbf{m}$, the posterior is given by

$$
\begin{aligned}
q(\mathbf{z}_s \mid \mathbf{z}_t, \mathbf{x}) &= \frac{q(\mathbf{z}_s|\mathbf{x})}{q(\mathbf{z}_t|\mathbf{x})}q(\mathbf{z}_t|\mathbf{z}_s) \\
&= \frac{1 - \gamma_s}{1 - \gamma_t}. \tag{16}
\end{aligned}
$$

Similarly, if $\mathbf{z}_s \neq \mathbf{m}$, the posterior is given by

$$
\begin{aligned}
q(\mathbf{z}_s \mid \mathbf{z}_t, \mathbf{x}) &= \frac{q(\mathbf{z}_s|\mathbf{x})}{q(\mathbf{z}_t|\mathbf{x})} q(\mathbf{z}_t|\mathbf{z}_s) \\
&= \frac{\gamma_s(\rho_s \mathbf{1}(\mathbf{z}_s = \mathbf{x}) + (1 - \rho_s)\frac{1}{K})}{1 - \gamma_t} \left(1 - \frac{\gamma_t}{\gamma_s}\right).
\end{aligned}
\tag{17}
$$

Combining (16) and (17), we obtain:

$$
\begin{aligned}
q(\mathbf{z}_s \mid \mathbf{z}_t, \mathbf{x}) &= \mathbf{1}(\mathbf{z}_s = \mathbf{m})\frac{1 - \gamma_s}{1 - \gamma_t} + \mathbf{1}(\mathbf{z}_s \neq \mathbf{m})\frac{\gamma_s(\rho_s \mathbf{1}(\mathbf{z}_s = \mathbf{x}) + (1 - \rho_s)\frac{1}{K})}{1 - \gamma_t}\left(1 - \frac{\gamma_t}{\gamma_s}\right) \\
&= \mathbf{1}(\mathbf{z}_s = \mathbf{m})\frac{1 - \gamma_s}{1 - \gamma_t} + \mathbf{1}(\mathbf{z}_s \neq \mathbf{m})\frac{\gamma_s - \gamma_t}{1 - \gamma_t}\left(\rho_s \mathbf{1}(\mathbf{z}_s = \mathbf{x}) + (1 - \rho_s)\frac{1}{K}\right).
\end{aligned}
\tag{18}
$$

Since $\mathbf{z}_s$, $\mathbf{z}_t$, $\mathbf{m}$ and $\mathbf{x}$ are all one-hot vectors, we replace the indicator functions with inner products to get (7). This step ensures valid gradient propagation when we replace $\mathbf{x}$ with $\mathbf{x}_\theta$, which is not a one-hot vector, in the backward process.

## B.2. Validity of Backward Process

In this section, we show that the model parameterization in (8) defines a valid density $p_\theta(\cdot \mid \mathbf{z}_t)$ for any $\mathbf{z}_t$.

*Lemma* 6. The model parameterization in (8) defines a valid density $p_\theta(\cdot \mid \mathbf{z}_t)$ for any $\mathbf{z}_t$.

*Proof.* We write $\mathbf{x}_\theta = \sum_{i=1}^{K} c_i e_i$ where $e_i$ is the $i$-th $K + 1$-dimensional canonical basis vector, and $c_i = \mathbf{x}_\theta^\top e_i$. We have $c = (c_1, ..., c_K, 0) \in \Delta_{K+1}$.

**Case 1: $\mathbf{z}_t = \mathbf{m}$.** In this case, the backward process is given by:

$$
p_\theta(\mathbf{z}_s \mid \mathbf{z}_t) = (1 - \mathbf{z}_s^\top \mathbf{m})\frac{\gamma_s - \gamma_t}{1 - \gamma_t}(\rho_s \mathbf{z}_s^\top \mathbf{x}_\theta + (1 - \rho_s)\frac{1}{K}) + \mathbf{z}_s^\top \mathbf{m}\frac{1 - \gamma_s}{1 - \gamma_t}
\tag{19}
$$

Summing over the support of $p_\theta$ in (19) gives:

$$
\begin{aligned}
\sum_{\mathbf{z}_s} p_\theta(\mathbf{z}_s \mid \mathbf{z}_t) &= \sum_{\mathbf{z}_s}\left[(1 - \mathbf{z}_s^\top \mathbf{m})\frac{\gamma_s - \gamma_t}{1 - \gamma_t}(\rho_s \mathbf{z}_s^\top \mathbf{x}_\theta + (1 - \rho_s)\frac{1}{K}) + \mathbf{z}_s^\top \mathbf{m}\frac{1 - \gamma_s}{1 - \gamma_t}\right] \\
&\stackrel{(i)}{=} \frac{1 - \gamma_s}{1 - \gamma_t} + \frac{\gamma_s - \gamma_t}{1 - \gamma_t}\left(\rho_s \sum_{\mathbf{z}_s} \mathbf{z}_s \mathbf{x}_\theta + 1 - \rho_s\right) \\
&= \frac{1 - \gamma_s}{1 - \gamma_t} + \frac{\gamma_s - \gamma_t}{1 - \gamma_t} \\
&= 1,
\end{aligned}
\tag{20}
$$

where $(i)$ is from the fact that $\mathbf{x}_\theta$ is a probability vector over the vocabulary, and hence $\sum_{\mathbf{z}_s} \mathbf{z}_s^\top \mathbf{x}_\theta = 1$. Therefore, $p_\theta(\cdot|\mathbf{z}_t)$ defines a probability density in this case.

**Case 2: $\mathbf{z}_t \neq \mathbf{m}$.** In this case, we have $\mathbf{z}_s \neq \mathbf{m}$ since [MASK] is absorbing. Using the fact $\frac{a_1+a_2}{b_1+b_2} = \frac{b_1}{b_1+b_2}\frac{a_1}{b_1} + \frac{b_2}{b_1+b_2}\frac{a_2}{b_2}$, we obtain:

$$
\begin{aligned}
\sum_{\mathbf{z}_s} p_\theta(\mathbf{z}_s \mid \mathbf{z}_t) &= \sum_{\mathbf{z}_s \neq \mathbf{m}} \frac{\rho_s \mathbf{z}_s^\top \mathbf{x}_\theta + (1-\rho_s)\frac{1}{K}}{\rho_t \mathbf{z}_t^\top \mathbf{x}_\theta + (1-\rho_t)\frac{1}{K}} \left( \frac{\rho_t}{\rho_s}\mathbf{z}_s^\top \mathbf{z}_t + \frac{\rho_s - \rho_t}{\rho_s}\frac{1}{K} \right) \\
&= \sum_{\mathbf{z}_s \neq \mathbf{m}} \frac{\sum_i c_i(\rho_s \mathbf{z}_s^\top e_i + (1-\rho_s)\frac{1}{K})}{\sum_i c_i(\rho_t \mathbf{z}_t^\top e_i + (1-\rho_t)\frac{1}{K})} \left( \frac{\rho_t}{\rho_s}\mathbf{z}_s^\top \mathbf{z}_t + \frac{\rho_s - \rho_t}{\rho_s}\frac{1}{K} \right) \\
&= \sum_{\mathbf{z}_s \neq \mathbf{m}} \sum_i \underbrace{\left( \frac{c_i(\rho_t \mathbf{z}_t^\top e_i + (1-\rho_t)\frac{1}{K})}{\sum_i c_i(\rho_t \mathbf{z}_t^\top e_i + (1-\rho_t)\frac{1}{K})} \right)}_{\kappa_i} \frac{\rho_s \mathbf{z}_s^\top e_i + (1-\rho_s)\frac{1}{K}}{\rho_t \mathbf{z}_t^\top e_i + (1-\rho_t)\frac{1}{K}} \left( \frac{\rho_t}{\rho_s}\mathbf{z}_s^\top \mathbf{z}_t + \frac{\rho_s - \rho_t}{\rho_s}\frac{1}{K} \right) \\
&= \sum_i \kappa_i \sum_{\mathbf{z}_s \neq \mathbf{m}} \frac{\rho_s \mathbf{z}_s^\top e_i + (1-\rho_s)\frac{1}{K}}{\rho_t \mathbf{z}_t^\top e_i + (1-\rho_t)\frac{1}{K}} \left( \frac{\rho_t}{\rho_s}\mathbf{z}_s^\top \mathbf{z}_t + \frac{\rho_s - \rho_t}{\rho_s}\frac{1}{K} \right) \\
&= \sum_i \kappa_i = 1
\end{aligned}
$$

Thus, $p_\theta(\cdot|\mathbf{z}_t)$ defines a probability density in this case. $\qquad\square$

## B.3. Derivation of Continuous-time Backward Transition Rate

In this section, we derive the transition rate for the backward process defined in (8) similarly to the forward transition rate (6). For each $t \in (0, 1]$ and $\mathbf{z}_t, \mathbf{z}_s \in \mathcal{V}$, define

$$
\tilde{Q}_t^\theta(\mathbf{z}_s, \mathbf{z}_t) = \begin{cases}
-\dfrac{1}{K}\dfrac{\rho_t'}{\rho_t}\dfrac{\rho_t \mathbf{z}_s^\top \mathbf{x}_\theta(\mathbf{z}_t, t) + (1-\rho_t)\frac{1}{K}}{\rho_t \mathbf{z}_t^\top \mathbf{x}_\theta(\mathbf{z}_t, t) + (1-\rho_t)\frac{1}{K}} & \text{if } \mathbf{z}_s \neq \mathbf{z}_t,\ \mathbf{z}_t \neq \mathbf{m} \\[2ex]
\dfrac{1}{K}\dfrac{\rho_t'}{\rho_t}\dfrac{1 - (\rho_t \mathbf{z}_t^\top \mathbf{x}_\theta(\mathbf{z}_t, t) + (1-\rho_t)\frac{1}{K})}{\rho_t \mathbf{z}_t^\top \mathbf{x}_\theta(\mathbf{z}_t, t) + (1-\rho_t)\frac{1}{K}} & \text{if } \mathbf{z}_s = \mathbf{z}_t,\ \mathbf{z}_t \neq \mathbf{m} \\[2ex]
-\dfrac{\gamma_t'}{1-\gamma_t}\left( \rho_t \mathbf{z}_s^\top \mathbf{x}_\theta(\mathbf{z}_t, t) + (1-\rho_t)\frac{1}{K} \right) & \text{if } \mathbf{z}_s \neq \mathbf{z}_t,\ \mathbf{z}_t = \mathbf{m} \\[2ex]
\dfrac{\gamma_t'}{1-\gamma_t} & \text{if } \mathbf{z}_t = \mathbf{z}_s = \mathbf{m} \\[2ex]
0 & \text{if } \mathbf{z}_t \neq \mathbf{m},\ \mathbf{z}_s = \mathbf{m}
\end{cases}
\tag{21}
$$

*Proposition* 7. Let $\tilde{Q}_t^\theta$ be defined as in (21). Then the following statements hold:

- $\tilde{Q}_t^\theta$ is a valid infinitesimal generator in the sense of Gat et al. (2024); in particular,

$$
\sum_{\mathbf{z}_s} \tilde{Q}_t^\theta(\mathbf{z}_s, \mathbf{z}_t) = 0, \qquad \tilde{Q}_t^\theta(\mathbf{z}_s, \mathbf{z}_t) \geq 0, \forall\, \mathbf{z}_s \neq \mathbf{z}_t.
$$

- The discrete-time backward process defined in (8) coincide with $\tilde{Q}_t^\theta$. In particular, for adjacent time points $s < t$ with $\Delta t = t - s$, we have

$$
p_\theta(\mathbf{z}_s \mid \mathbf{z}_t) = \delta_{\mathbf{z}_t, \mathbf{z}_s} + \Delta t\, \tilde{Q}_t^\theta(\mathbf{z}_s, \mathbf{z}_t) + o(\Delta t).
$$

*Proof.* The proof follows the same argument as in Lemma 2. First, we show that $\tilde{Q}_t^\theta$ is a valid generator.

Fix an arbitrary $\mathbf{z}_t$. Since $\rho_t',\ \gamma_t' \leq 0$, all off-diagonal entries of $\tilde{Q}_t^\theta$ are non-negative, i.e.,

$$
\tilde{Q}_t^\theta(\mathbf{z}_s, \mathbf{z}_t) \geq 0, \qquad \forall\, \mathbf{z}_s \neq \mathbf{z}_t.
$$

Moreover, by construction of $\tilde{Q}_t^\theta$ we have the row-sum condition

$$
\sum_{\mathbf{z}_s} \tilde{Q}_t^\theta(\mathbf{z}_s, \mathbf{z}_t) = 0.
$$

Therefore, $\tilde{Q}_t^\theta$ is a valid infinitesimal generator in the sense of Gat et al. (2024).

Next, we show that $\tilde{Q}_t^\theta$ is indeed the backward transition kernel of the backward process (8). Let $s = t - \Delta t$ with $\Delta t > 0$. Then by first-order Taylor expansion we have

$$\rho_s = \rho_t - \rho_t' \Delta t + o(\Delta t), \qquad \gamma_s = \gamma_t - \gamma_t' \Delta t + o(\Delta t). \tag{22}$$

We now verify that the discrete backward kernel (7) satisfies

$$p_\theta(\mathbf{z}_s \mid \mathbf{z}_t) = \delta_{\mathbf{z}_s, \mathbf{z}_t} + \Delta t \, \tilde{Q}_t^\theta(\mathbf{z}_s, \mathbf{z}_t) + o(\Delta t)$$

case by case.

**Case 1. $\mathbf{z}_t, \mathbf{z}_s \neq \mathbf{m}$, $\mathbf{z}_s \neq \mathbf{z}_t$.** In this case, $\mathbf{z}_s^\top \mathbf{m} = 0$ and $\mathbf{z}_s^\top \mathbf{z}_t = 0$. Thus, we have

$$
\begin{aligned}
p_\theta(\mathbf{z}_s \mid \mathbf{z}_t) &= \frac{\rho_s \, \mathbf{z}_s^\top \mathbf{x}_\theta + (1 - \rho_s)\frac{1}{K}}{\rho_t \, \mathbf{z}_t^\top \mathbf{x}_\theta + (1 - \rho_t)\frac{1}{K}} \left( \frac{\rho_s - \rho_t}{\rho_s} \frac{1}{K} \right) \\
&= \left( \frac{\rho_t \, \mathbf{z}_s^\top \mathbf{x}_\theta + (1 - \rho_t)\frac{1}{K}}{\rho_t \, \mathbf{z}_t^\top \mathbf{x}_\theta + (1 - \rho_t)\frac{1}{K}} - \frac{\rho_t'(\frac{1}{K} - \mathbf{z}_s^\top \mathbf{x}_\theta)}{\rho_t \, \mathbf{z}_t^\top \mathbf{x}_\theta + (1 - \rho_t)\frac{1}{K}} \Delta t + o(\Delta t) \right) \left( \frac{\rho_t'}{\rho_t}(-\frac{1}{K})\Delta t + o(\Delta t) \right) \\
&= \Delta t \left( -\frac{\rho_t'}{\rho_t} \frac{1}{K} \frac{\rho_t \, \mathbf{z}_s^\top \mathbf{x}_\theta + (1 - \rho_t)\frac{1}{K}}{\rho_t \, \mathbf{z}_t^\top \mathbf{x}_\theta + (1 - \rho_t)\frac{1}{K}} \right) + o(\Delta t).
\end{aligned}
\tag{23}
$$

**Case 2. $\mathbf{z}_t = \mathbf{z}_s \neq \mathbf{m}$.** We have

$$
\begin{aligned}
p_\theta(\mathbf{z}_s \mid \mathbf{z}_t) &= \frac{\rho_s \, \mathbf{z}_t^\top \mathbf{x}_\theta + (1 - \rho_s)\frac{1}{K}}{\rho_t \, \mathbf{z}_t^\top \mathbf{x}_\theta + (1 - \rho_t)\frac{1}{K}} \left( \frac{\rho_t}{\rho_s} + \frac{\rho_s - \rho_t}{\rho_s} \frac{1}{K} \right) \\
&= \left( 1 + \frac{\rho_t'(\frac{1}{K} - \mathbf{z}_t^\top \mathbf{x}_\theta)}{\rho_t \, \mathbf{z}_t^\top \mathbf{x}_\theta + (1 - \rho_t)\frac{1}{K}} \Delta t + o(\Delta t) \right) \left( 1 + \frac{\rho_t'}{\rho_t}(1 - \frac{1}{K})\Delta t + o(\Delta t) \right) \\
&= 1 + \Delta t \left( \frac{\rho_t'(\frac{1}{K} - \mathbf{z}_t^\top \mathbf{x}_\theta)}{\rho_t \, \mathbf{z}_t^\top \mathbf{x}_\theta + (1 - \rho_t)\frac{1}{K}} + \frac{\rho_t'}{\rho_t}(1 - \frac{1}{K}) \right) + o(\Delta t) \\
&= 1 + \Delta t \frac{\rho_t'}{\rho_t} \frac{1}{K} \frac{1 - (\rho_t \, \mathbf{z}_t^\top \mathbf{x}_\theta + (1 - \rho_t)\frac{1}{K})}{\rho_t \, \mathbf{z}_t^\top \mathbf{x}_\theta + (1 - \rho_t)\frac{1}{K}} + o(\Delta t).
\end{aligned}
\tag{24}
$$

**Case 3. $\mathbf{z}_s \neq \mathbf{m}$, $\mathbf{z}_t = \mathbf{m}$.** We have

$$
\begin{aligned}
p_\theta(\mathbf{z}_s \mid \mathbf{z}_t) &= \frac{\gamma_s - \gamma_t}{1 - \gamma_t} \left( \rho_s \, \mathbf{z}_s^\top \mathbf{x}_\theta + (1 - \rho_s)\frac{1}{K} \right) \\
&= -\Delta t \frac{\gamma_t'}{1 - \gamma_t} \left( \rho_t \, \mathbf{z}_s^\top \mathbf{x}_\theta + (1 - \rho_t)\frac{1}{K} + \rho_t'(\frac{1}{K} - \mathbf{z}_s^\top \mathbf{x}_\theta)\Delta t \right) \\
&= -\Delta t \frac{\gamma_t'}{1 - \gamma_t} \left( \rho_t \, \mathbf{z}_s^\top \mathbf{x}_\theta + (1 - \rho_t)\frac{1}{K} \right) + o(\Delta t).
\end{aligned}
\tag{25}
$$

**Case 4. $\mathbf{z}_s = \mathbf{z}_t = \mathbf{m}$.** We have

$$p_\theta(\mathbf{z}_s \mid \mathbf{z}_t) = \frac{1 - \gamma_s}{1 - \gamma_t} = 1 + \Delta t \frac{\gamma_t'}{1 - \gamma_t} + o(\Delta t). \tag{26}$$

**Case 5: $\mathbf{z}_t \neq \mathbf{m}$, $\mathbf{z}_s = \mathbf{m}$.** The last case holds from the fact that SCDD doesn't allow remasking.

In all cases, $p_\theta(\mathbf{z}_s \mid \mathbf{z}_t)$ aligns with the process implied by $\tilde{Q}_t^\theta(\mathbf{z}_t, \mathbf{z}_s)$ in (21), which completes the proof. $\qquad \square$

## C. Analysis of Training Loss

### C.1. Discrete-time NELBO

Following standard variational arguments on diffusion models (Sohl-Dickstein et al., 2015; Sahoo et al., 2024), we start from the classical ELBO:

$$
\begin{aligned}
\mathbb{E}[-\log p_\theta(\mathbf{x})] &\leq \mathbb{E}_q\left[-\log p_\theta(\mathbf{x}, \mathbf{z}_{0:1}) + \log q(\mathbf{z}_{0:1} \mid \mathbf{x})\right] \\
&= \underbrace{-\mathbb{E}_{q(\mathbf{x}, \mathbf{z}_{0:1})}\left[\log p_\theta(\mathbf{x} \mid \mathbf{z}_0)\right]}_{\mathcal{L}_{\text{reconstruction}}^T} + \underbrace{\mathbb{E}_{q(\mathbf{x}, \mathbf{z}_{0:1})}\left[D_{\text{KL}}\big(q(\mathbf{z}_1 \mid \mathbf{x})\|p_\theta(\mathbf{z}_1)\big)\right]}_{\mathcal{L}_{\text{prior}}} \\
&\quad + \underbrace{\mathbb{E}_{q(\mathbf{x}, \mathbf{z}_{0:1})}\left[\sum_{i=1}^T D_{\text{KL}}\big(q(\mathbf{z}_{t_{i-1}} \mid \mathbf{z}_{t_i}, \mathbf{x})\|p_\theta(\mathbf{z}_{t_{i-1}} \mid \mathbf{z}_{t_i}))\big)\right]}_{\mathcal{L}_{\text{diffusion}}^T} =: \mathcal{L}_{\text{NELBO}}^T.
\end{aligned} \tag{27}
$$

In this section, we explicitly calculate the the diffusion loss $\mathcal{L}_{\text{diffusion}}^T$. As before, we use $t, s$ to denote two adjacent time points with $t > s$. We first rewrite the summation over $i$ as an expectation over all time points:

$$
\begin{aligned}
\mathcal{L}_{\text{diffusion}}^T &= \mathbb{E}_q\left[\sum_{i=1}^T D_{\text{KL}}\big(q(\mathbf{z}_{t_{i-1}} \mid \mathbf{z}_{t_i}, \mathbf{x})\|p_\theta(\mathbf{z}_{t_{i-1}} \mid \mathbf{z}_{t_i}))\big)\right] \\
&= \frac{1}{T}\sum_{i=1}^T \mathbb{E}_q\left[T D_{\text{KL}}\big(q(\mathbf{z}_{t_{i-1}} \mid \mathbf{z}_{t_i}, \mathbf{x})\|p_\theta(\mathbf{z}_{t_{i-1}} \mid \mathbf{z}_{t_i}))\big)\right] \\
&= \mathbb{E}_{t \sim \mathcal{U}\{t_1, \ldots, t_T\}}\mathbb{E}_q\left[T D_{\text{KL}}\big(q(\mathbf{z}_s \mid \mathbf{z}_t, \mathbf{x})\|p_\theta(\mathbf{z}_s \mid \mathbf{z}_t))\big)\right]
\end{aligned} \tag{28}
$$

Since

$$
\mathcal{D}_{\text{KL}}(q\|p_\theta) = \mathbb{E}_q[\log q] - \mathbb{E}_q[\log p_\theta],
$$

the first (negative entropy) term in KL-divergence is independent of $\theta$, and minimizing $\mathcal{L}_{\text{diffusion}}^T$ is equivalent to minimizing the expectation of second cross entropy term. With a little abuse of notation, we drop the first term and redefine the $\mathcal{L}_{\text{diffusion}}^T$ as

$$
\mathcal{L}_{\text{diffusion}}^T = -\mathbb{E}_{t \sim \mathcal{U}\{t_1, \ldots, t_T\}}\mathbb{E}_q\left[T \log p_\theta(\mathbf{z}_s \mid \mathbf{z}_t)\right] \tag{29}
$$

When $\mathbf{z}_s = \mathbf{m}$, we have $\mathbf{z}_t = \mathbf{m}$ and $p_\theta(\mathbf{z}_s \mid \mathbf{z}_t) = \frac{1-\gamma_s}{1-\gamma_t}$. This case is independent of $\theta$ and will be ignored in optimization. We therefore restrict to the case $\mathbf{z}_s \neq \mathbf{m}$.

**Case 1: $\mathbf{z}_t \neq \mathbf{m}$.** Substituting $p_\theta(\mathbf{z}_s \mid \mathbf{z}_t)$ yields

$$
\log p_\theta(\mathbf{z}_s \mid \mathbf{z}_t) = \log\left(\frac{\rho_s \mathbf{z}_s^\top \mathbf{x}_\theta + (1-\rho_s)\frac{1}{K}}{\rho_t \mathbf{z}_t^\top \mathbf{x}_\theta + (1-\rho_t)\frac{1}{K}}\right) + \log\left(\frac{\rho_t}{\rho_s}\mathbf{z}_s^\top \mathbf{z}_t + \frac{\rho_s - \rho_t}{\rho_s}\frac{1}{K}\right)
$$

The second term in $\log p_\theta(\mathbf{z}_s \mid \mathbf{z}_t)$ is independent of $\theta$ and drops from the loss. Thus, up to an additive constant,

$$
\begin{aligned}
\mathcal{L}_{\text{diffusion}}^T \mid_{\mathbf{z}_t \neq \mathbf{m}} &= -\mathbb{E}_{t \sim \mathcal{U}\{t_1, \ldots, t_T\}}\mathbb{E}_q\left[T\sum_{\mathbf{z}_s \neq \mathbf{m}} \frac{\rho_s \mathbf{z}_s^\top \mathbf{x} + (1-\rho_s)\frac{1}{K}}{\rho_t \mathbf{z}_t^\top \mathbf{x} + (1-\rho_t)\frac{1}{K}}\left(\frac{\rho_t}{\rho_s}\mathbf{z}_s^\top \mathbf{z}_t + \frac{\rho_s - \rho_t}{\rho_s}\frac{1}{K}\right)\log\frac{\rho_s \mathbf{z}_s^\top \mathbf{x}_\theta + (1-\rho_s)\frac{1}{K}}{\rho_t \mathbf{z}_t^\top \mathbf{x}_\theta + (1-\rho_t)\frac{1}{K}}\right] \\
&= -\mathbb{E}_{t \sim \mathcal{U}\{t_1, \ldots, t_T\}}\mathbb{E}_q\left[T\sum_{\mathbf{v} \neq \mathbf{m}} \frac{\rho_s \mathbf{v}^\top \mathbf{x} + (1-\rho_s)\frac{1}{K}}{\rho_t \mathbf{z}_t^\top \mathbf{x} + (1-\rho_t)\frac{1}{K}}\left(\frac{\rho_t}{\rho_s}\mathbf{v}^\top \mathbf{z}_t + \frac{\rho_s - \rho_t}{\rho_s}\frac{1}{K}\right)\log\frac{\rho_s \mathbf{v}^\top \mathbf{x}_\theta + (1-\rho_s)\frac{1}{K}}{\rho_t \mathbf{z}_t^\top \mathbf{x}_\theta + (1-\rho_t)\frac{1}{K}}\right],
\end{aligned} \tag{30}
$$

up to some $\theta$-independent additive constant. We replaced $\mathbf{z}_s$ with $\mathbf{v}$ in the last line to make it clear that the summation doesn't depend on the value of $\mathbf{z}_s$.

**Case 2: $\mathbf{z}_t = \mathbf{m}$.** In this case,

$$\log p_\theta(\mathbf{z}_s \mid \mathbf{z}_t = \mathbf{m}) = \log\left(\rho_s \mathbf{z}_s^\top \mathbf{x}_\theta + (1 - \rho_s)\tfrac{1}{K}\right) + \log\left(\tfrac{\gamma_s - \gamma_t}{1 - \gamma_t}\right),$$

Dropping the second term in $\log p_\theta(\mathbf{z}_s \mid \mathbf{z}_t)$ and substituting into the cross-entropy objective yields

$$\mathcal{L}_{\text{diffusion}}^T \mid_{\mathbf{z}_t} = -\mathbb{E}_{t \sim \mathcal{U}\{t_1, \ldots, t_T\}} \mathbb{E}_q \left[ T \sum_{\mathbf{z}_s \neq \mathbf{m}} \frac{\gamma_s(\rho_s \mathbf{z}_s^\top \mathbf{x} + (1 - \rho_s)\frac{1}{K})}{1 - \gamma_t} \frac{\gamma_s - \gamma_t}{\gamma_s} \log\left(\rho_s \mathbf{z}_s^\top \mathbf{x}_\theta + (1 - \rho_s)\tfrac{1}{K}\right) \right]$$

$$= -\mathbb{E}_{t \sim \mathcal{U}\{t_1, \ldots, t_T\}} \mathbb{E}_q \left[ T \sum_{\mathbf{v} \neq \mathbf{m}} \frac{\gamma_s - \gamma_t}{1 - \gamma_t} (\rho_s \mathbf{v}^\top \mathbf{x} + (1 - \rho_s)\tfrac{1}{K}) \log(\rho_s \mathbf{v}^\top \mathbf{x}_\theta + (1 - \rho_s)\tfrac{1}{K}) \right], \quad (31)$$

up to some $\theta$-independent additive constant.

### C.2. Continuous-time NELBO

Next, we show that reconstruction loss and prior loss vanishes as $T \to \infty$, and explicitly derive $\mathcal{L}_{\text{diffusion}}^\infty$.

**Reconstruction Loss** As $T \to \infty$, $\rho_0, \gamma_0 \to 1^-$ by the design of noising schedule. Therefore, we have

$$\mathbf{z}_0 \sim \text{Cat}\left(\cdot; \gamma_0\left(\rho_0 \mathbf{x} + (1 - \rho_0)\mathbf{u}\right) + (1 - \gamma_0)\mathbf{m}\right)$$

$$\stackrel{d}{\Longrightarrow} \text{Cat}\left(\cdot; \mathbf{x}\right), \quad (32)$$

where "$\stackrel{d}{\Longrightarrow}$" stands for "converge in distribution". Therefore, we have the reconstruction loss vanishes as follows:

$$\lim_{T \to \infty} \mathcal{L}_{\text{reconstruction}}^T = \lim_{T \to \infty} \mathbb{E}_q[-\log p_\theta(\mathbf{x} \mid \mathbf{z}_0)]$$

$$\stackrel{(i)}{=} \lim_{T \to \infty} \mathbb{E}_q \left[ -\log \frac{\rho_{-1}\mathbf{x}^\top \mathbf{x}_\theta(\mathbf{z}_0, 0) + (1 - \rho_{-1})\frac{1}{K}}{\rho_0 \mathbf{z}_0^\top \mathbf{x}_\theta(\mathbf{z}_0, 0) + (1 - \rho_0)\frac{1}{K}} \left( \frac{\rho_0}{\rho_{-1}} \mathbf{x}^\top \mathbf{z}_0 + \frac{\rho_{-1} - \rho_0}{\rho_{-1}} \frac{1}{K} \right) \right]$$

$$= \mathbb{E}_q \left[ -\log \frac{\mathbf{x}^\top \mathbf{x}_\theta(\mathbf{x}, 0)}{\mathbf{x}^\top \mathbf{x}_\theta(\mathbf{x}, 0)} \left( \mathbf{x}^\top \mathbf{x} \right) \right]$$

$$= \mathbb{E}_q[-\log 1] = 0, \quad (33)$$

where $(i)$ is from plugging $\mathbf{z}_t = \mathbf{z}_0, t = 0$ into (8).

**Prior Loss** At $t = 1$, the forward marginal distribution becomes $q(\mathbf{z}_1 \mid \mathbf{x}) = \mathbf{1}(\mathbf{z}_1 = \mathbf{m})$, indicating that the forward noising process ultimately transforms clean data $\mathbf{x}$ to $\mathbf{m}$ at $t = 1$. We also choose the prior to be $p_\theta(\mathbf{z}_1) = q(\mathbf{z}_1 \mid \mathbf{x}) = \mathbf{1}(\mathbf{z}_1 = \mathbf{m})$ so that the backward process starts from $\mathbf{m}$. Therefore, we have $\mathcal{L}_{\text{prior}}^T \equiv 0$.

**Diffusion Loss** In this section, we discuss the continuous-time version of the loss function in (11). We assume that $\rho_t$ and $\gamma_t$ are decreasing, differentiable functions of time.

*Proposition* 8. Pick adjacent time points $s < t$ and let $\Delta t = t - s$ be the time step of the above discrete stochastic process. Then as $\Delta t \to 0$ or, equivalently, $T \to \infty$, we have

$$\mathcal{L}_{\text{diffusion}}^\infty = \begin{cases} \mathbb{E}_{t \sim \mathcal{U}[0,1]} \mathbb{E}_q \left[ \sum_{\mathbf{v} \neq \mathbf{z}_t, \mathbf{m}} \left( \frac{(\rho_t \mathbf{v}^\top \mathbf{x} + (1 - \rho_t)\frac{1}{K})\frac{\rho_t'}{\rho_t}\frac{1}{K}}{\rho_t \mathbf{z}_t^\top \mathbf{x} + (1 - \rho_t)\frac{1}{K}} \log \frac{\rho_t \mathbf{v}^\top \mathbf{x}_\theta + (1 - \rho_t)\frac{1}{K}}{\rho_t \mathbf{z}_t^\top \mathbf{x}_\theta + (1 - \rho_t)\frac{1}{K}} \right) - \frac{\rho_t'(-\mathbf{z}_t^\top \mathbf{x}_\theta + \frac{1}{K})}{\rho_t \mathbf{z}_t^\top \mathbf{x}_\theta + (1 - \rho_t)\frac{1}{K}} \right], & \text{if } \mathbf{z}_t \neq \mathbf{m}, \\[4mm] \mathbb{E}_{t \sim \mathcal{U}[0,1]} \mathbb{E}_q \left[ \sum_{\mathbf{v} \neq \mathbf{m}} \frac{\gamma_t'}{1 - \gamma_t} (\rho_t \mathbf{v}^\top \mathbf{x} + (1 - \rho_t)\frac{1}{K}) \log(\rho_t \mathbf{v}^\top \mathbf{x}_\theta + (1 - \rho_t)\frac{1}{K}) \right], & \text{if } \mathbf{z}_t = \mathbf{m}. \end{cases}$$

$$(34)$$

*Proof.* We distinguish two cases of $\mathbf{z}_t$ to calculate the continuous-time loss function.

**Case 1: $\mathbf{z}_t \neq \mathbf{m}$.** If $\mathbf{z}_t \neq \mathbf{m}$, using the first-order Taylor expansion of $\rho_s$ and $\gamma_s$, we have

$$\rho_s \mathbf{z}_s^\top \mathbf{x} + (1 - \rho_s)\tfrac{1}{K} = (\rho_t - \rho_t' \Delta t)\mathbf{z}_s^\top \mathbf{x} + (1 - \rho_t + \rho_t' \Delta t)\tfrac{1}{K} + o(\Delta t)$$
$$= (\rho_t \mathbf{z}_s^\top \mathbf{x} + (1 - \rho_t)\tfrac{1}{K}) + \Delta t \rho_t'(-\mathbf{z}_s^\top \mathbf{x} + \tfrac{1}{K}) + o(\Delta t),$$
$$\log \frac{\rho_s \mathbf{z}_s^\top \mathbf{x}_\theta + (1 - \rho_s)\tfrac{1}{K}}{\rho_t \mathbf{z}_t^\top \mathbf{x}_\theta + (1 - \rho_t)\tfrac{1}{K}} = \log \frac{\rho_t \mathbf{z}_s^\top \mathbf{x}_\theta + (1 - \rho_t)\tfrac{1}{K}}{\rho_t \mathbf{z}_t^\top \mathbf{x}_\theta + (1 - \rho_t)\tfrac{1}{K}} + \Delta t \frac{\rho_t'(-\mathbf{z}_s^\top \mathbf{x}_\theta + \tfrac{1}{K})}{\rho_t \mathbf{z}_s^\top \mathbf{x}_\theta + (1 - \rho_t)\tfrac{1}{K}} + o(\Delta t),$$
$$\frac{\rho_t}{\rho_s}\mathbf{z}_s^\top \mathbf{z}_t + \frac{\rho_s - \rho_t}{\rho_s}\tfrac{1}{K} = \mathbf{z}_s^\top \mathbf{z}_t + \frac{\rho_t'}{\rho_t}\Delta t(\mathbf{z}_s^\top \mathbf{z}_t - \tfrac{1}{K}) + o(\Delta t). \tag{35}$$

Substituting the above expansions into the discrete loss (30),

$$\mathcal{L}_{\text{diffusion}}^T \mid_{\mathbf{z}_t \neq \mathbf{m}} = -T\mathbb{E}_{t \sim \mathcal{U}\{t_1,\ldots,t_T\}}\mathbb{E}_q \sum_{\mathbf{z}_s \neq \mathbf{m}} \left[ \mathbf{z}_s^\top \mathbf{z}_t \frac{\rho_t \mathbf{z}_s^\top \mathbf{x} + (1 - \rho_t)\tfrac{1}{K}}{\rho_t \mathbf{z}_t^\top \mathbf{x} + (1 - \rho_t)\tfrac{1}{K}} \left( \log \frac{\rho_t \mathbf{z}_s^\top \mathbf{x}_\theta + (1 - \rho_t)\tfrac{1}{K}}{\rho_t \mathbf{z}_t^\top \mathbf{x}_\theta + (1 - \rho_t)\tfrac{1}{K}} + \Delta t \frac{\rho_t'(-\mathbf{z}_s^\top \mathbf{x}_\theta + \tfrac{1}{K})}{\rho_t \mathbf{z}_s^\top \mathbf{x}_\theta + (1 - \rho_t)\tfrac{1}{K}} \right) \right.$$
$$\left. + \Delta t \frac{\rho_t'(-\mathbf{z}_s^\top \mathbf{x} + \tfrac{1}{K})\mathbf{z}_s^\top \mathbf{z}_t + (\rho_t \mathbf{z}_s^\top \mathbf{x} + (1 - \rho_t)\tfrac{1}{K})\frac{\rho_t'}{\rho_t}(\mathbf{z}_s^\top \mathbf{z}_t - \tfrac{1}{K})}{\rho_t \mathbf{z}_t^\top \mathbf{x} + (1 - \rho_t)\tfrac{1}{K}} \log \frac{\rho_t \mathbf{z}_s^\top \mathbf{x}_\theta + (1 - \rho_t)\tfrac{1}{K}}{\rho_t \mathbf{z}_t^\top \mathbf{x}_\theta + (1 - \rho_t)\tfrac{1}{K}} + o(\Delta t) \right].$$

We distinguish two cases when doing summation over $\mathbf{z}_s$ to simplify the above expansion: (i) if $\mathbf{z}_s = \mathbf{z}_t$, all logarithmic terms vanish and the loss reduces to

$$-T\Delta t\mathbb{E}_{t \sim \mathcal{U}\{t_1,\ldots,t_T\}}\mathbb{E}_q \left[ \frac{\rho_t'(-\mathbf{z}_t^\top \mathbf{x}_\theta + \tfrac{1}{K})}{\rho_t \mathbf{z}_t^\top \mathbf{x}_\theta + (1 - \rho_t)\tfrac{1}{K}} \right] + o(1) = -\mathbb{E}_{t \sim \mathcal{U}\{t_1,\ldots,t_T\}}\mathbb{E}_q \left[ \frac{\rho_t'(-\mathbf{z}_t^\top \mathbf{x}_\theta + \tfrac{1}{K})}{\rho_t \mathbf{z}_t^\top \mathbf{x}_\theta + (1 - \rho_t)\tfrac{1}{K}} \right] + o(1);$$

(ii) if $\mathbf{z}_s \neq \mathbf{z}_t$, the first term inside of expectation and summation vanishes and we obtain

$$\mathbb{E}_{t \sim \mathcal{U}\{t_1,\ldots,t_T\}}\mathbb{E}_q \sum_{\mathbf{z}_s \neq \mathbf{z}_t, \mathbf{m}} \left[ \frac{(\rho_t \mathbf{z}_s^\top \mathbf{x} + (1 - \rho_t)\tfrac{1}{K})\frac{\rho_t'}{\rho_t}\tfrac{1}{K}}{\rho_t \mathbf{z}_t^\top \mathbf{x} + (1 - \rho_t)\tfrac{1}{K}} \log \frac{\rho_t \mathbf{z}_s^\top \mathbf{x}_\theta + (1 - \rho_t)\tfrac{1}{K}}{\rho_t \mathbf{z}_t^\top \mathbf{x}_\theta + (1 - \rho_t)\tfrac{1}{K}} \right] + o(1). \tag{36}$$

Combining the two cases, letting $T \to \infty$, and replacing the $\mathbf{z}_s$ notation with $\mathbf{v}$ yields the desired limit for $\mathbf{z}_t \neq \mathbf{m}$:

$$\mathcal{L}_{\text{diffusion}}^\infty \mid_{\mathbf{z}_t \neq \mathbf{m}} = \mathbb{E}_{t \sim \mathcal{U}[0,1]}\mathbb{E}_q \left[ \sum_{\mathbf{v} \neq \mathbf{z}_t, \mathbf{m}} \left( \frac{(\rho_t \mathbf{v}^\top \mathbf{x} + (1 - \rho_t)\tfrac{1}{K})\frac{\rho_t'}{\rho_t}\tfrac{1}{K}}{\rho_t \mathbf{z}_t^\top \mathbf{x} + (1 - \rho_t)\tfrac{1}{K}} \log \frac{\rho_t \mathbf{v}^\top \mathbf{x}_\theta + (1 - \rho_t)\tfrac{1}{K}}{\rho_t \mathbf{z}_t^\top \mathbf{x}_\theta + (1 - \rho_t)\tfrac{1}{K}} \right) - \frac{\rho_t'(-\mathbf{z}_t^\top \mathbf{x}_\theta + \tfrac{1}{K})}{\rho_t \mathbf{z}_t^\top \mathbf{x}_\theta + (1 - \rho_t)\tfrac{1}{K}} \right].$$

**Case 2: $\mathbf{z}_t = \mathbf{m}$.** Using the first-order Taylor expansions to get

$$\frac{\gamma_t - \gamma_s}{1 - \gamma_t} = \Delta t \frac{\gamma_t'}{1 - \gamma_t},$$
$$\rho_s \mathbf{z}_s^\top \mathbf{x} + (1 - \rho_s)\tfrac{1}{K} = (\rho_t \mathbf{z}_s^\top \mathbf{x} + (1 - \rho_t)\tfrac{1}{K}) + \Delta t \rho_t'(-\mathbf{z}_s^\top \mathbf{x} + \tfrac{1}{K}) + o(\Delta t),$$
$$\log(\rho_s \mathbf{z}_s^\top \mathbf{x}_\theta + (1 - \rho_s)\tfrac{1}{K}) = \log(\rho_t \mathbf{z}_s^\top \mathbf{x}_\theta + (1 - \rho_t)\tfrac{1}{K}) + \Delta t \frac{\rho_t'(-\mathbf{z}_s^\top \mathbf{x}_\theta + \tfrac{1}{K})}{\rho_t \mathbf{z}_s^\top \mathbf{x}_\theta + (1 - \rho_t)\tfrac{1}{K}} + o(\Delta t). \tag{37}$$

Substituting into the discrete loss (31) yields

$$\mathcal{L}_{\text{diffusion}}^\infty \mid_{\mathbf{z}_t = \mathbf{m}} = \mathbb{E}_{t \sim \mathcal{U}[0,1]}\mathbb{E}_q \left[ \sum_{\mathbf{z}_s \neq \mathbf{m}} \frac{\gamma_t'}{1 - \gamma_t}(\rho_t \mathbf{z}_s^\top \mathbf{x} + (1 - \rho_t)\tfrac{1}{K}) \log(\rho_t \mathbf{z}_s^\top \mathbf{x}_\theta + (1 - \rho_t)\tfrac{1}{K}) \right], \tag{38}$$

which completes the proof.

$\square$

# D. Connections Between SCDD and Other Models

## D.1. SCDD and MDLM

In this section, we discuss the relation between SCDD and MDLM (Sahoo et al., 2024), and show SCDD can be reduced to MDLM under specific parameter settings. Firstly, recall several facts of MDLM (see Sahoo et al. (2024) for full derivations):

**Marginal Distribution.** The marginal distribution at time $t \in [0,1]$ is

$$q(\mathbf{z}_t | \mathbf{x}) = \text{Cat}(\mathbf{z}_t; \alpha_t^{\text{MDLM}} \mathbf{x} + (1 - \alpha_t^{\text{MDLM}}) \mathbf{m}), \tag{39}$$

where $\alpha_t^{\text{MDLM}} \in [0, 1]$.

**Forward Process.**

$$q(\mathbf{z}_t \mid \mathbf{z}_s) = \text{Cat}\left(\mathbf{z}_t; \alpha_{t|s}^{\text{MDLM}} \mathbf{z}_s + (1 - \alpha_{t|s}^{\text{MDLM}}) \mathbf{m}\right), \tag{40}$$

where $\alpha_{t|s} = \alpha_t / \alpha_s$.

**Backward Process.**

$$p_\theta\left(\mathbf{z}_s \mid \mathbf{z}_t\right) \begin{cases} \text{Cat}\left(\mathbf{z}_s; \mathbf{z}_t\right), & \mathbf{z}_t \neq \mathbf{m}, \\ \text{Cat}\left(\mathbf{z}_s; \dfrac{(1-\alpha_s)\mathbf{m} + \left(\alpha_s^{\text{MDLM}} - \alpha_t^{\text{MDLM}}\right) \mathbf{x}_\theta(\mathbf{z}_t, t)}{1 - \alpha_t^{\text{MDLM}}}\right). & \mathbf{z}_t = \mathbf{m}, \end{cases} \tag{41}$$

**Training Loss.**

$$\mathcal{L}_{\text{diffusion}}^T = -T \mathbb{E}_t \mathbb{E}_q \left[\frac{\alpha_s^{\text{MDLM}} - \alpha_t^{\text{MDLM}}}{1 - \alpha_t^{\text{MDLM}}} \log(\mathbf{x}_\theta^\top \mathbf{x})\right], \qquad \text{discrete version} \tag{42}$$

$$\mathcal{L}_{\text{diffusion}}^\infty = \mathbb{E}_t \mathbb{E}_q \left[\frac{(\alpha_t^{\text{MDLM}})'}{1 - \alpha_t^{\text{MDLM}}} \log(\mathbf{x}_\theta^\top \mathbf{x})\right], \qquad \text{continuous version} \tag{43}$$

*Proposition* 9. If we set $\rho_t \equiv 1$ and $\gamma_t \equiv \alpha_t^{\text{MDLM}}$ in (2), SCDD can recover MDLM. In particular, the marginal distribution, forward process & posterior, and loss function can recover MDLM.

*Proof.* It's easy to see the marginal (2) becomes (39) and the forward process (3) becomes (40) whenever $\rho_t \equiv 1$ and $\gamma_t \equiv \alpha_t^{\text{MDLM}}$. Therefore, the backward process of MDLM and SCDD also align based on Bayes' Rule and the same model parameterization. Finally, we verify the training loss. If $\mathbf{z}_t = \mathbf{m}$, the diffusion loss (31) becomes

$$\mathcal{L}_{\text{diffusion}}^T \mid_{\mathbf{z}_t = \mathbf{m}} = -T \mathbb{E}_t \mathbb{E}_q \sum_{\mathbf{v} \neq \mathbf{m}} \frac{\gamma_s - \gamma_t}{1 - \gamma_t} (\rho_s \mathbf{v}^\top \mathbf{x} + (1 - \rho_s) \tfrac{1}{K}) \log(\rho_t \mathbf{v}^\top \mathbf{x}_\theta + (1 - \rho_t) \tfrac{1}{K})$$

$$= -T \mathbb{E}_t \mathbb{E}_q \left[\frac{\gamma_s - \gamma_t}{1 - \gamma_t} \log \mathbf{x}_\theta^\top \mathbf{x}\right].$$

If $\mathbf{z}_t \neq \mathbf{m}$, then $\mathbf{z}_t = \mathbf{z}_s = \mathbf{x}$. From (30), we have

$$\mathcal{L}_{\text{diffusion}}^T \mid_{\mathbf{z}_t = \mathbf{x}} = -T \mathbb{E}_t \mathbb{E}_q \left[\frac{\mathbf{x}^\top \mathbf{z}_s}{\mathbf{x}^\top \mathbf{z}_t} (\mathbf{z}_s^\top \mathbf{z}_t) \log \frac{\mathbf{z}_s^\top \mathbf{x}_\theta}{\mathbf{z}_t^\top \mathbf{x}_\theta}\right]$$

$$= -T \log 1 = 0.$$

Therefore, we recover (42).

For the continuous loss, if $\mathbf{z}_t \neq \mathbf{m}$, (34) vanishes since $\rho_t' \equiv 0$. If $\mathbf{z}_t = \mathbf{m}$, substituting $\rho_t \equiv 1$ yields

$$\mathcal{L}_{\text{diffusion}}^\infty \mid_{\mathbf{z}_t = \mathbf{m}} = \mathbb{E}_t \mathbb{E}_q \left[\frac{\gamma_t'}{1 - \gamma_t} \log(\mathbf{x}^\top \mathbf{x}_\theta)\right],$$

which coincides with (43). $\qquad \square$

## D.2. SCDD and GIDD

In this section, we discuss the relation between the SCDD and GIDD (von Rütte et al., 2025), and derive the noise schedule of SCDD that recovers the marginal distribution of GIDD. Firstly, recall several facts of GIDD (see von Rütte et al. (2025) for full derivations):

**Marginal Distribution.**   The marginal distribution at time $t \in [0, 1]$ is

$$q_t^{\text{GIDD}}(\mathbf{z}_t \mid \mathbf{x}) = \text{Cat}\left(\mathbf{z}_t; \alpha_t^{\text{GIDD}}\mathbf{x} + \beta_t^{\text{GIDD}}\pi_t\right), \tag{44}$$

where $\beta_t^{\text{GIDD}} = 1 - \alpha_t^{\text{GIDD}}$ and $\alpha_t^{\text{GIDD}} \in [0, 1]$.

**Forward Process.**   Given a mixing rate $\alpha_t$ and a time-dependent mixing distribution $\pi_t$, the forward Markov chain is defined such that the cumulative transition from $s$ to $t$ satisfies

$$q_{t|s}^{\text{GIDD}}(\mathbf{z}_t \mid \mathbf{z}_s) = \text{Cat}\left(\mathbf{z}_t; \alpha_{t|s}^{\text{GIDD}}\mathbf{z}_s + \beta_{t|s}^{\text{GIDD}}\pi_{t|s}\right), \tag{45}$$

where

$$\alpha_{t|s}^{\text{GIDD}} = \frac{\alpha_t^{\text{GIDD}}}{\alpha_s^{\text{GIDD}}}, \qquad \beta_{t|s}^{\text{GIDD}}\pi_{t|s} = \beta_t^{\text{GIDD}}\pi_t - \frac{\alpha_t^{\text{GIDD}}}{\alpha_s^{\text{GIDD}}}\beta_s^{\text{GIDD}}\pi_s. \tag{46}$$

**Forward Transition Rate**   The forward rate matrix of GIDD is given by

$$R_t^{\text{GIDD}}(\mathbf{z}_s, \mathbf{z}_t) = \frac{(\alpha_t^{\text{GIDD}})'}{\alpha_t^{\text{GIDD}}}\delta_{\mathbf{z}_s, \mathbf{z}_t} + \mathbf{z}_t^{\top}(\beta_t^{\text{GIDD}}\pi_t' - \frac{(\alpha_t^{\text{GIDD}})'}{\alpha_t^{\text{GIDD}}}\pi_t), \tag{47}$$

and thus

$$q_{t|s}^{\text{GIDD}}(\mathbf{z}_t|\mathbf{z}_s) = \delta_{\mathbf{z}_s, \mathbf{z}_t} + R_t^{\text{GIDD}}(\mathbf{z}_s, \mathbf{z}_t)\Delta t + o(\Delta t),$$

where $\Delta t = t - s$.

**Backward process.**   The backward process is parameterized as

$$p_\theta^{\text{GIDD}}(\mathbf{z}_s \mid \mathbf{z}_t) = \frac{q_{t|s}^{\text{GIDD}}(\mathbf{z}_t \mid \mathbf{z}_s)q_s^{\text{GIDD}}(\mathbf{z}_s \mid \mathbf{x}_\theta)}{q_t^{\text{GIDD}}(\mathbf{z}_t \mid \mathbf{x}_\theta)}. \tag{48}$$

**Training Loss.**   The continuous-time negative ELBO is given by

$$-\log q(x) \le \mathbb{E}_{t,\mathbf{z}_t}\left[w_t(\mathbf{z}_t, x)\left(D_{\text{KL}}(q_t^{\text{GIDD}}(\cdot \mid \mathbf{x})\|q_t^{\text{GIDD}}(\cdot \mid \mathbf{x}_\theta)) + D_{\text{IS}}(q_t^{\text{GIDD}}(\mathbf{z}_t \mid \mathbf{x})\|q_t^{\text{GIDD}}(\mathbf{z}_t \mid \mathbf{x}_\theta)))\right]\right] + C, \tag{49}$$

with weighting term

$$w_t(\mathbf{z}_t, \mathbf{x}) = \frac{1}{q_t^{\text{GIDD}}(\mathbf{z}_t \mid \mathbf{x})}\mathbf{z}_t^{\top}\left(\beta_t^{\text{GIDD}}\pi_t' - \frac{(\alpha_t^{\text{GIDD}})'}{\alpha_t^{\text{GIDD}}}\pi_t\right), \tag{50}$$

where $D_{\text{IS}}$ denotes the Itakura–Saito divergence and $C$ is the ELBO constant.

The next proposition shows that GIDD can be rewritten under our marginal parameterization, but with a non-absorbing [MASK] token and more coupled forward transition kernel.

*Proposition* 10 (Reparameterization of GIDD).   Consider the forward transition kernel

$$q(\mathbf{z}_t|\mathbf{z}_s) = \text{Cat}\left(\mathbf{z}_t; \gamma_{t|s}\rho_{t|s}\mathbf{z}_s + \gamma_t(1 - \rho_{t|s})\mathbf{u} + \left((1 - \gamma_t) - \gamma_{t|s}\rho_{t|s}(1 - \gamma_s)\right)\mathbf{m}\right). \tag{51}$$

This kernel induces the marginal distribution in (2). Its continuous-time transition rate is

$$R_t(\mathbf{z}_t, \mathbf{z}_s) = \left(\frac{\rho_t'}{\rho_t} + \frac{\gamma_t'}{\gamma_t}\right)\delta_{\mathbf{z}_s, \mathbf{z}_t} - \mathbf{z}_t^{\top}\left[\gamma_t\frac{\rho_t'}{\rho_t}\mathbf{u} + \left((1 - \gamma_t)\frac{\rho_t'}{\rho_t} + \frac{\gamma_t'}{\gamma_t}\right)\mathbf{m}\right]. \tag{52}$$

Moreover, under the reparameterization

$$\begin{cases} \alpha_t^{\mathrm{GIDD}} = \rho_t \gamma_t, \\ \beta_t^{\mathrm{GIDD}} = 1 - \alpha_t^{\mathrm{GIDD}}, \\ \beta_t^{\mathrm{GIDD}} \pi_t = \gamma_t (1 - \rho_t) \mathbf{u} + (1 - \gamma_t) \mathbf{m}, \end{cases} \tag{53}$$

the forward process induced by (51) coincides with the GIDD forward process in (45). Consequently, the marginal distribution and continuous-time transition rate induced by (51) coincide with the GIDD marginal distribution and transition rate in (44) and (47), respectively.

*Remark* 11. Proposition 10 shows that the forward kernel in (51) provides a reparameterized form of the GIDD forward process under the parameter translation in (53). This kernel induces the same marginal distribution as (2), but unlike the SCDD forward kernel, it does not impose the absorbing-mask condition $q(\mathbf{z}_t = \mathbf{m} \mid \mathbf{z}_s = \mathbf{m}) = 1$. Therefore, SCDD and GIDD can be expressed under a common marginal parameterization, while differing in their forward transition kernels. This distinction explains why the uniform-transition and masking components are coupled in the GIDD transition rate, whereas they are controlled separately in SCDD.

*Proof.* We prove the proposition in three steps.

**Step 1: Marginal distribution.** We first verify that the transition kernel in (51) induces the marginal distribution in (2). Suppose that at time $s$,

$$q(\mathbf{z}_s \mid \mathbf{x}) = \mathrm{Cat}\Big(\mathbf{z}_s; \gamma_s\big(\rho_s \mathbf{x} + (1 - \rho_s)\mathbf{u}\big) + (1 - \gamma_s)\mathbf{m}\Big).$$

By the forward transition kernel (51), the marginal at time $t$ has probability vector

$$\gamma_{t|s}\rho_{t|s}\Big[\gamma_s\big(\rho_s \mathbf{x} + (1 - \rho_s)\mathbf{u}\big) + (1 - \gamma_s)\mathbf{m}\Big] + \gamma_t(1 - \rho_{t|s})\mathbf{u}$$
$$+ \Big((1 - \gamma_t) - \gamma_{t|s}\rho_{t|s}(1 - \gamma_s)\Big)\mathbf{m}. \tag{54}$$

Using $\gamma_{t|s} = \gamma_t/\gamma_s$ and $\rho_{t|s} = \rho_t/\rho_s$, the coefficient of $\mathbf{x}$ is

$$\gamma_{t|s}\rho_{t|s}\gamma_s\rho_s = \gamma_t\rho_t.$$

The coefficient of $\mathbf{u}$ is

$$\gamma_{t|s}\rho_{t|s}\gamma_s(1 - \rho_s) + \gamma_t(1 - \rho_{t|s}) = \gamma_t\rho_t\frac{1 - \rho_s}{\rho_s} + \gamma_t\left(1 - \frac{\rho_t}{\rho_s}\right)$$
$$= \gamma_t(1 - \rho_t). \tag{55}$$

The coefficient of $\mathbf{m}$ is

$$\gamma_{t|s}\rho_{t|s}(1 - \gamma_s) + (1 - \gamma_t) - \gamma_{t|s}\rho_{t|s}(1 - \gamma_s) = 1 - \gamma_t.$$

Therefore,

$$q(\mathbf{z}_t \mid \mathbf{x}) = \mathrm{Cat}\Big(\mathbf{z}_t; \gamma_t\big(\rho_t \mathbf{x} + (1 - \rho_t)\mathbf{u}\big) + (1 - \gamma_t)\mathbf{m}\Big),$$

which is exactly (2).

**Step 2: Forward transition rate.** Let $t = s + \Delta t$. From (51), the probability that $\mathbf{z}_t$ retains $\mathbf{z}_s$ is

$$\gamma_{t|s}\rho_{t|s} = \frac{\gamma_t\rho_t}{\gamma_s\rho_s}.$$

Using first-order Taylor expansions,

$$\frac{\gamma_t}{\gamma_s} = 1 + \frac{\gamma_t'}{\gamma_t}\Delta t + o(\Delta t), \qquad \frac{\rho_t}{\rho_s} = 1 + \frac{\rho_t'}{\rho_t}\Delta t + o(\Delta t),$$

we obtain

$$\gamma_{t|s}\rho_{t|s} = 1 + \left(\frac{\gamma_t'}{\gamma_t} + \frac{\rho_t'}{\rho_t}\right)\Delta t + o(\Delta t).$$

Similarly, the coefficient of the uniform transition component is

$$\gamma_t(1 - \rho_{t|s}) = -\gamma_t \frac{\rho'_t}{\rho_t} \Delta t + o(\Delta t),$$

and the coefficient of the masking component is

$$(1 - \gamma_t) - \gamma_{t|s}\rho_{t|s}(1 - \gamma_s) = -\left[(1 - \gamma_t)\frac{\rho'_t}{\rho_t} + \frac{\gamma'_t}{\gamma_t}\right]\Delta t + o(\Delta t). \tag{56}$$

Thus,

$$q(\mathbf{z}_t \mid \mathbf{z}_s) = \delta_{\mathbf{z}_s, \mathbf{z}_t} + \Delta t\, R_t(\mathbf{z}_t, \mathbf{z}_s) + o(\Delta t),$$

where

$$R_t(\mathbf{z}_t, \mathbf{z}_s) = \left(\frac{\rho'_t}{\rho_t} + \frac{\gamma'_t}{\gamma_t}\right)\delta_{\mathbf{z}_s, \mathbf{z}_t} - \mathbf{z}_t^\top \left[\gamma_t \frac{\rho'_t}{\rho_t}\mathbf{u} + \left((1 - \gamma_t)\frac{\rho'_t}{\rho_t} + \frac{\gamma'_t}{\gamma_t}\right)\mathbf{m}\right],$$

which proves (52).

**Step 3: Equivalence with GIDD.** It remains to connect the forward process in (51) with the original GIDD parameterization. Under the reparameterization

$$\alpha_t^{\mathrm{GIDD}} = \rho_t\gamma_t, \qquad \beta_t^{\mathrm{GIDD}} = 1 - \alpha_t^{\mathrm{GIDD}}, \qquad \beta_t^{\mathrm{GIDD}}\pi_t = \gamma_t(1 - \rho_t)\mathbf{u} + (1 - \gamma_t)\mathbf{m},$$

we have

$$\alpha_{t|s}^{\mathrm{GIDD}} = \frac{\alpha_t^{\mathrm{GIDD}}}{\alpha_s^{\mathrm{GIDD}}} = \frac{\rho_t\gamma_t}{\rho_s\gamma_s} = \rho_{t|s}\gamma_{t|s}.$$

Moreover,

$$\begin{aligned}
\beta_t^{\mathrm{GIDD}}\pi_t - \alpha_{t|s}^{\mathrm{GIDD}}\beta_s^{\mathrm{GIDD}}\pi_s &= \gamma_t(1 - \rho_t)\mathbf{u} + (1 - \gamma_t)\mathbf{m} \\
&\quad - \gamma_{t|s}\rho_{t|s}\Big[\gamma_s(1 - \rho_s)\mathbf{u} + (1 - \gamma_s)\mathbf{m}\Big] \\
&= \gamma_t(1 - \rho_{t|s})\mathbf{u} + \big((1 - \gamma_t) - \gamma_{t|s}\rho_{t|s}(1 - \gamma_s)\big)\mathbf{m}. \tag{57}
\end{aligned}$$

Therefore, substituting (53) into the GIDD forward kernel (45) gives

$$\alpha_{t|s}^{\mathrm{GIDD}}\mathbf{z}_s + \beta_t^{\mathrm{GIDD}}\pi_t - \alpha_{t|s}^{\mathrm{GIDD}}\beta_s^{\mathrm{GIDD}}\pi_s = \gamma_{t|s}\rho_{t|s}\mathbf{z}_s + \gamma_t(1 - \rho_{t|s})\mathbf{u} + \big((1 - \gamma_t) - \gamma_{t|s}\rho_{t|s}(1 - \gamma_s)\big)\mathbf{m},$$

which is exactly the probability vector in (51). Hence, the forward process induced by (51) coincides with the GIDD forward process. Consequently, the marginal distribution and continuous-time transition rate induced by (51) coincide with the GIDD marginal distribution and transition rate in (44) and (47), respectively. □

*Remark* 12. [Comparison of Forward Transition Rates between SCDD and GIDD.]

Under CTMC theory and using the same parameterization of marginal distribution, the GIDD's forward transition rate is given by:

$$R_t^{\mathrm{GIDD}}(\mathbf{z}_s, \mathbf{z}_t) = \left(\frac{\gamma'_t}{\gamma_t} + \frac{\rho'_t}{\rho_t}\right)\mathbf{z}_s^\top \mathbf{z}_t - \mathbf{z}_t^\top \left(\gamma_t \frac{\rho'_t}{\rho_t}\mathbf{u} + \left((1 - \gamma_t)\frac{\rho'_t}{\rho_t} + \frac{\gamma'_t}{\gamma_t}\right)\mathbf{m}\right),$$

and SCDD's transition rate, when $\mathbf{z}_s \neq \mathbf{m}$, is given by equation (6):

$$R_t^{\mathrm{SCDD}}(\mathbf{z}_s, \mathbf{z}_t) = \left(\frac{\gamma'_t}{\gamma_t} + \frac{\rho'_t}{\rho_t}\right)\mathbf{z}_s^\top \mathbf{z}_t - \mathbf{z}_t^\top \left(\frac{\rho'_t}{\rho_t}\mathbf{u} + \frac{\gamma'_t}{\gamma_t}\mathbf{m}\right).$$

Our formulation yields a simpler forward process by making $\mathbf{m}$ an absorbing state. In particular, the uniform transition noise $\mathbf{u}$ and the absorbing mask noise $\mathbf{m}$ become **decoupled**, and their rates are independently controlled by the marginal parameters $\rho_t$ and $\gamma_t$, respectively. In contrast, GIDD couples these two noise channels, making the parameters' respective effects less explicit. This decoupling makes the forward dynamics more interpretable and substantially reduces the algebraic complexity of the backward formulas and the resulting training loss, which in turn simplifies implementation and lowers maintenance cost in practice.

# E. Experimental Details

## E.1. Training Details.

For fair comparison, we retrained MDLM (Sahoo et al., 2024) and GIDD (von Rütte et al., 2025) along with our SCDD model. The detailed hyper-parameter setting for different datasets can be found in Table 7. We train MDLM and SCDD with a diffusion process and loss function defined over $T = 1000$ discrete time steps, and train GIDD using the continuous-time loss with dynamic weighting to guarantee the best performance, see (von Rütte et al., 2025) for the impact of weighting function on GIDD training. We also align the marginal distribution of SCDD with GIDD to guarantee two models see distributionally equivalent samples during training. Both of the models are trained using a noise schedule that achieves uniform transition peak ratio of $p_u \in \{0.1, 0.2\}$ at $t = 0.5$. See Appendix E.4 for detailed discussion.

All the models use the DiT (Peebles & Xie, 2023) structure as the backbone of denoising networks, and GPT-2 tokenizer (Radford et al., 2019). Following MDLM and GIDD, we use the SMALL variant of DiT as the denoising network with 12 Transformer blocks, 12 attention heads per block, and 768 hidden dimension, yielding 166M trainable parameters. Time conditioning is projected to a dimension of 128 before being injected into the network. No dropout is applied during training.

For LM1B (Chelba et al., 2013) dataset, all the model are trained with a batch size of 512, context length of 128, for 500k steps, yielding 33B ($512 \times 128 \times 500k$) training tokens in total. For OWT (Gokaslan & Cohen, 2019) dataset, all the models are trained with batch size of 256, context length of 512, for 1M steps, yielding 131B ($256 \times 512 \times 1M$) training tokens in total. We leave the last 100k docs as validation as in MDLM training. For ablation study, we train the model on Wikitext-103 (Merity et al., 2017) with a batch size of 128, context length 512, for 100k steps, yielding 7B ($128 \times 512 \times 100k$) training tokens in total.

For optimization, we use the AdamW optimizer (Kingma & Ba, 2014) with hyperparameters $\beta_1 = 0.9$, $\beta_2 = 0.999$, and $\epsilon = 1e-9$. The learning rate increases linearly from $1 \times 10^{-6}$ to the peak rate $5 \times 10^{-4}$ over the first 10k steps, then follows a cosine-decay schedule to $5 \times 10^{-5}$ for the remainder of the training. To stabilize training, we use `bfloat16` mixed precision and maintain an Exponential Moving Average of model parameters with a decay rate of 0.9999. Training are completed on a single node of 4 (Wikitext-103) or 8 (LM1B and OWT) NVIDIA H100-80GB GPUs.

*Table 7.* Hyper-parameter configuration for each dataset.

|  | Wikitext-103 | LM1B | OWT |
|---|---|---|---|
| Context Length | 512 | 128 | 512 |
| Training Steps $N$ | 100K | 500K | 1M |
| Batch Size | 128 | 512 | 256 |
| Number of GPUs | 4 | 8 | 8 |
| Learning Rate | 5e-4 | | |
| T | 1000 | | |
| Training and testing hardware | H100-80GB GPUs | | |

## E.2. Unconditional Text Generation Details.

As noted by Zheng et al. (2025), 32-bit floating-point Gumbel-max categorical sampling suffers from numerical precision issues. Therefore, we use 64-bit floating-point for all text generations. ReMDM-cap and ReMDM-conf (Wang et al., 2026) are two customized samplers that can be readily applied to pretrained MDLM model to elicit self-correction. For ReMDM-cap we use the default $\eta_{cap} = 0.01$. We apply nucleus sampling (Holtzman et al., 2020) with $p = 0.9$ throughout the experiments as it improves the quality of generated texts.

**Entropy Results.** We report the unigram entropy results in Table 8 in addition to the generative perplexity results as a sanity check.

**LLM-as-a-judge.** In the LLM-as-a-judge experiment, we borrow the LLM prompt from von Rütte et al. (2025), as in Figure 4.

Beyond the matched-schedule setting, we also evaluate SCDD and GIDD+ under a cross-ratio setting by comparing the

*Table 8.* Entropy on LM1B and OWT datasets across sampling steps. Two decimal places are shown.

| ENTROPY | LM1B (STEPS) | | | | | OWT (STEPS) | | | | | |
|---|---|---|---|---|---|---|---|---|---|---|---|
| | 16 | 32 | 64 | 128 | 256 | 32 | 64 | 128 | 256 | 512 | 1024 |
| MDLM | 4.37 | 4.36 | 4.36 | 4.36 | 4.36 | 5.26 | 5.23 | 5.21 | 5.19 | 5.19 | 5.17 |
| REMDM CAP 0.01 | 4.37 | 4.36 | 4.36 | 4.36 | 4.35 | 5.26 | 5.23 | 5.19 | 5.17 | 5.15 | 5.13 |
| REMDM CONFIDENCE | 4.37 | 4.36 | 4.35 | 4.35 | 4.36 | 5.26 | 5.22 | 5.19 | 5.17 | 5.15 | 5.13 |
| GIDD ($p_u = 0.1$) | 4.34 | 4.35 | 4.35 | 4.36 | 4.36 | 5.09 | 5.07 | 5.06 | 5.06 | 5.05 | 5.05 |
| GIDD ($p_u = 0.2$) | 4.34 | 4.35 | 4.36 | 4.36 | 4.36 | 5.09 | 5.08 | 5.07 | 5.07 | 5.06 | 5.06 |
| SCDD ($p_u = 0.1$) | 4.21 | 4.23 | 4.24 | 4.24 | 4.24 | 4.83 | 4.86 | 4.86 | 4.86 | 4.86 | 4.85 |
| SCDD ($p_u = 0.2$) | 4.22 | 4.24 | 4.24 | 4.25 | 4.25 | 4.85 | 4.87 | 4.86 | 4.87 | 4.86 | 4.85 |

**Evaluation Prompt**

```
1.  Clarity and coherence:  Keeping in mind that the text may be cut off in the
beginning and at the end due to it being an excerpt, how clear and understandable
is the text?
2.  Grammaticality:  Are there any grammatical errors in the text?
3.  Factuality:  If applicable, is the factually verifiable information stated in
the text (e.g.  facts about geography, history, etc.)  accurate and reliable?
4.  Writing style:  How well is the text written in terms of style and fluency?  Do
the sentences flow well, is the vocabulary appropriate?
5.  Creativity:  How original and creative is the text?
For each category, give a short justification before providing the final score.
Your answer should be following the JSON format, with one top-level key for each
aspect ('clarity', 'grammaticality', 'factuality', 'style', and 'creativity').
Each aspect, in turn, should be a JSON object consisting of a 'reasoning' and
'score' key in that order.  The 'reasoning' key should contain a short justification
for the score, and the 'score' key should contain the score itself.
Please keep the following in mind:  - Give your justification first before deciding
on a final score.  - Only output the JSON containing the justifications and scores
and nothing else.  - Keep in mind that the presented paragraph may be an excerpt
from a longer document, so it may not be fully self-contained.  Do not deduct points
for issues arising from this.
The text to be graded is as follows:  ''' text '''
```

*Figure 4.* Evaluation Prompt used for LLM-as-a-judge experiment.

best-performing SCDD ($p_u = 0.2$) with the best GIDD+ ($p_u = 0.1$). In this setup, clean OWT sequences are still corrupted using SCDD's forward noising process; consequently, GIDD+ ($p_u = 0.1$) must reconstruct inputs that are "dirtier" than those seen during its original training. Table 9 shows that SCDD consistently maintains statistically significant better performance in *Clarity*, *Factuality*, and *Style* across all sampling steps. Despite GIDD+ showing an advantage in *Creativity*, SCDD achieves a higher overall win rate. These results confirm that GIDD+ ($p_u = 0.1$), which is trained at a lower noise level, lacks the capacity to effectively correct texts at higher corruption levels.

### E.3. Benchmark Performance.

For each zero-shot task, we rank candidate completions by estimating the conditional log-likelihood of the answer $y$ given the question $x$. Because diffusion models are non-autoregressive, we score each candidate using the ELBO of the concatenated sequence $[x; y]$. Since $P(x)$ is identical across candidates, maximizing $\log P(x, y)$ is equivalent to maximizing $\log P(y \mid x)$.

Concretely, for each $[x; y]$ pair, we compute its contribution to the diffusion variational loss, which is stochastic due to sampling $t \sim \mathcal{U}(0, 1)$, and the corresponding forward noising process. We therefore use a Monte Carlo estimate, averaging the ELBO over $N = 10$ independent forward passes, and select the candidate with the highest mean ELBO. At last, we calculate the accuracy of selection. Results are reported in Table 10.

Since each of zero-shot tasks ranks answer candidates by their estimated ELBO, there is no self-correction during evaluation,

*Table 9.* Cross-ratio Setting — SCDD ($p_u$=0.2) vs GIDD+ ($p_u$=0.1). Values are formatted as SCDD (GIDD+). Significance: $^*p < 0.05$, $^{**}p < 0.01$.

| METRICS | STEPS | | | | | |
|---|---|---|---|---|---|---|
| | 32 | 64 | 128 | 256 | 512 | 1024 |
| CLARITY | 1.64 (1.52)$^*$ | 1.66 (1.51)$^*$ | 1.70 (1.46)$^{**}$ | 1.70 (1.54)$^{**}$ | 1.73 (1.54)$^{**}$ | 1.73 (1.52)$^{**}$ |
| GRAMM. | 1.39 (1.48) | 1.45 (1.51) | 1.49 (1.44) | 1.52 (1.51) | 1.54 (1.53) | 1.51 (1.49) |
| FACT. | 2.25 (2.13)$^{**}$ | 2.16 (2.07)$^*$ | 2.16 (2.01)$^{**}$ | 2.22 (2.08)$^{**}$ | 2.20 (2.05)$^{**}$ | 2.12 (2.01)$^*$ |
| STYLE | 1.64 (1.54) | 1.65 (1.54)$^*$ | 1.67 (1.49)$^{**}$ | 1.67 (1.55)$^*$ | 1.71 (1.57)$^*$ | 1.72 (1.52)$^{**}$ |
| CREATIVITY | 2.86 (3.21)$^{**}$ | 2.95 (3.27)$^{**}$ | 2.91 (3.25)$^{**}$ | 2.93 (3.26)$^{**}$ | 2.98 (3.31)$^{**}$ | 2.98 (3.27)$^{**}$ |
| WIN RATE | 52.4% | 53.9% | 57.8%$^*$ | 56.0% | 55.1% | 58.0%$^*$ |

making these benchmarks a measure of zero-shot likelihood rather than language generation. Uniform-noise diffusion models (including both SCDD and GIDD+) typically exhibit worse likelihood but stronger abilities in self-correction, few-step sampling, and generation tasks (Sahoo et al., 2026). Our generative PPL results (Table 3) confirm that SCDD produces higher-quality text than MDLM/GIDD at equal sampling steps, demonstrating that worse likelihood estimation does not translate to weaker language generation.

*Table 10.* Zero-shot accuracy on various language modeling benchmarks. Comparison between MDLM, GIDD, and SCDD at different noise levels ($p_u$).

| MODEL | ARC-E | ARC-C | BOOLQ | HELLASWAG | OBQA | PIQA | WINOG | AVG. |
|---|---|---|---|---|---|---|---|---|
| MDLM | 27.90 | 21.16 | 47.25 | 27.76 | 19.20 | 52.01 | 50.43 | 35.10 |
| GIDD ($p_u = 0.1$) | 26.85 | 22.01 | 48.99 | 26.36 | 15.60 | 50.76 | 50.27 | 34.41 |
| GIDD ($p_u = 0.2$) | 26.43 | 21.92 | 48.29 | 26.55 | 15.80 | 51.09 | 50.51 | 34.37 |
| SCDD ($p_u = 0.1$, OURS) | 26.64 | 22.01 | 48.13 | 26.73 | 17.20 | 50.05 | 48.78 | 34.22 |
| SCDD ($p_u = 0.2$, OURS) | 26.52 | 24.16 | 46.64 | 26.20 | 20.40 | 49.67 | 49.01 | 34.66 |

### E.4. Noise Schedule of SCDD

von Rütte et al. (2025) define the following time-dependent rates to elicit self-correction:

$$\alpha_t^{\text{GIDD}} = \frac{1-t}{C_t}, \qquad \beta_t^{\text{GIDD}} \pi_t = \frac{t}{C_t}\mathbf{m} + \frac{c_t}{C_t}\mathbf{u}, \tag{58}$$

$$\text{where } c_t = Bt^{\gamma/2}(1-t)^{\gamma/2}, \qquad C_t = 1 + c_t, \qquad B, \gamma > 0$$

Notably, the marginal probability of uniform transition noise ratio is a symmetric function of $t$, vanishing at endpoints $t = 0, 1$ and attaining its maximum at $t = 1/2$. In von Rütte et al. (2025), this maximum ratio is denoted by $p_u$, and can be expressed as

$$p_u = \frac{c_{1/2}}{C_{1/2}} = \frac{c_{1/2}}{1 + c_{1/2}}.$$

If $p_u$ is fixed, we can solve for the corresponding constant $B$ as:

$$B = \frac{c_{1/2}}{(1/2)^\gamma} = 2^\gamma \frac{p_u}{1 - p_u}.$$

To ensure a fair comparison between SCDD and GIDD, we align the marginal distribution of SCDD with that of GIDD in our experimental evaluation. By applying the parameter translation in (53), we obtain:

$$\begin{cases} \rho_t \gamma_t & = \dfrac{1-t}{1+c_t}, \\ (1-\rho_t)\gamma_t & = \dfrac{c_t}{1+c_t}, \\ 1 - \gamma_t & = \dfrac{t}{1+c_t}. \end{cases}$$

Solve the above equations to obtain noise schedule for SCDD in experiments:

$$\begin{cases} \gamma_t = \frac{1+c_t-t}{1+c_t}, \\ \rho_t = \frac{1-t}{1+c_t-t}, \end{cases} \tag{59}$$

where $c_t := 2^\gamma \frac{p_u}{1-p_u} t^{\gamma/2}(1-t)^{\gamma/2}, p_u \in \{0.1, 0.2\}$.

In ablation study, we use a noise schedule that attains the maximum noise ratio at a general time point $t_{\text{peak}}$, where $B$ and $c_t$ are given as follows:

$$B = \frac{p_u}{1-p_u} \cdot \frac{1}{t_{\text{peak}}^{\gamma t_{\text{peak}}}(1-t_{\text{peak}})^{\gamma(1-t_{\text{peak}})}}.$$

$$c_t = Bt^{\gamma t_{\text{peak}}}(1-t)^{\gamma(1-t_{\text{peak}})} = \frac{p_u}{1-p_u} \cdot \frac{t^{\gamma t_{\text{peak}}}(1-t)^{\gamma(1-t_{\text{peak}})}}{t_{\text{peak}}^{\gamma t_{\text{peak}}}(1-t_{\text{peak}})^{\gamma(1-t_{\text{peak}})}}. \tag{60}$$

# F. Related Works

**Mask Diffusion Language Model (MDLM/MDM) (Sahoo et al., 2024; Shi et al., 2024)**  In vanilla masked diffusion models, the forward process is defined by a Markov chain of the form

$$q(\mathbf{z}_t \mid \mathbf{z}_s) = \alpha_{t|s} \mathbf{x} + (1 - \alpha_{t|s}) \mathbf{m},$$

which induces a marginal distribution

$$q(\mathbf{z}_t \mid \mathbf{x}) = \text{Cat}(\mathbf{z}_t; \alpha_t\mathbf{x} + (1 - \alpha_t)\mathbf{m}).$$

Masked diffusion models offer significant computational advantages and substantially improve upon earlier discrete flow-matching approaches. However, they lack an explicit self-correction mechanism. In particular, once a token is decoded during inference, it becomes fixed and cannot be revised in subsequent denoising steps. As a result, early prediction errors accumulate over the backward process and harms generation quality. SCDD can be viewed as a generalization of MDLM to support self-correction. We refer to Appendix D.1 for a formal comparison.

**Remasking Diffusion Model (ReMDM) (Wang et al., 2026)**  ReMDM introduces a collection of post-doc samplers that can be readily applied to pretrained MDLM models. To elicit self-correction, it adopts a *remasking* mechanism in the backward (inference) process:

$$q(\mathbf{z}_s \mid \mathbf{z}_t, \mathbf{x}) = \begin{cases} \text{Cat}(\mathbf{z}_s; (1 - \sigma_t)\mathbf{x} + \sigma_t\mathbf{m}), \\ \text{Cat}\left(\mathbf{z}_s; \frac{\alpha_s - (1-\sigma_t)\alpha_t}{1-\alpha_t}\mathbf{x} + \frac{1-\alpha_s-\sigma_t\alpha_t}{1-\alpha_t}\mathbf{m}\right), \end{cases}$$

where $\alpha_t, \sigma_t \in [0, 1]$ are noise schedules. The non-Markovian forward process is derived from Bayes' rule.

While this remasking-based formulation enables token revision, it exhibits two notable limitations. First, ReMDM performs self-correction through a non-[MASK] → [MASK] → non-[MASK] procedure, in which the intermediate masking step is redundant: it takes 2 steps to correct a token. This may reduce the parallel self-correction efficiency during inference. Second, the performance of ReMDM highly depends on the hyperparameters that affect the actual schedule of $\sigma_t$, which introduces additional tuning cost.

**Generalized Interpolating Discrete Diffusion (GIDD) (von Rütte et al., 2025)**  GIDD replaces the absorbing mask probability vector $\mathbf{m}$ in the marginal distribution of MDLM (Sahoo et al., 2024) with a more general corruption distribution $\pi_t$, defined as a mixture of absorbing mask $\mathbf{m}$ and uniform transition noise $\mathbf{u}$. The resulting marginal distribution takes the form

$$q(\mathbf{z}_t \mid \mathbf{x}) := \text{Cat}(\mathbf{z}_t; \alpha_t\mathbf{x} + (1 - \alpha_t)\pi_t),$$

which is induced by the following Markov transition:

$$q(\mathbf{z}_t \mid \mathbf{z}_s) := \text{Cat}(\mathbf{z}_t; \alpha_{t|s}\mathbf{z}_s + (1 - \alpha_t)\pi_t - \alpha_{t|s}(1 - \alpha_s)\pi_s).$$

SCDD and GIDD can both be viewed as generalizations of vanilla masked diffusion language models, differing primarily in their choices of the forward transition kernel. We refer to Proposition 10 and Remark 12 for a detailed comparison.

Compared to SCDD, GIDD exhibits the following limitations. First, GIDD controls uniform corruption and masking through an entangled parameterization, whereas SCDD decouples these two sources of noise. This decoupling leads to a substantially simpler backward process and a more tractable training loss. Second, GIDD doesn't eliminate the remasking step during inference, while SCDD supports direct token-to-token self-correction during both training and inference without introducing an intermediate masking stage.

**Informed Corrector (Zhao et al., 2024)**   Zhao et al. (2024) use continuous-time Markov chains (CTMCs) to describe the forward and backward processes:

$$q_{t|s}(\mathbf{y} \mid \mathbf{x}) = \delta_{\mathbf{x},\mathbf{y}} + R_t(\mathbf{x}, \mathbf{y})\Delta t + o(\Delta t),$$

where $\mathbf{x}, \mathbf{y} \in \mathcal{V}^D$ denote token sequences (e.g. articles or paragraphs), and

$$R_t(\mathbf{x}, \mathbf{y}) = \sum_{d=1}^{D} \beta_t R(\mathbf{x}_d, \mathbf{y}_d), \qquad R(\mathbf{x}_d, \mathbf{y}_d) := \begin{cases} 1 & \mathbf{y}_d = \mathbf{m}, \ \mathbf{x}_d \neq \mathbf{m}, \\ -1 & \mathbf{y}_d = \mathbf{x}_d \neq \mathbf{m}, \\ 0 & \mathbf{x}_d = \mathbf{y}_d = \mathbf{m}. \end{cases}$$

This formulation is equivalent to the forward Markov chain used in masked diffusion language models (MDLM), and the resulting marginal distribution can be written as

$$q_t(\mathbf{x}_{t,d} \mid \mathbf{x}_{0,d}) = \alpha_t \mathbf{1}(\mathbf{x}_{t,d} = \mathbf{x}_{0,d}) + (1 - \alpha_t)\mathbf{1}(\mathbf{x}_{t,d} = \mathbf{m}).$$

During inference, Zhao et al. (2024) introduce a *confidence score $c_d$* for each token position $\mathbf{x}_d$, defined for example as

$$c_d = \log p_\theta(\mathbf{x}_{0,d} \mid \mathbf{x}_t^{/d}) - \max_{i \neq d} \log p_\theta(\mathbf{x}_{0,i} \mid \mathbf{x}_t^{/d}),$$

where $p_\theta(\cdot \mid \mathbf{x}_t^{/d}) \approx q_{0|t}(\cdot \mid \mathbf{x}^{/d})$ denotes the denoising model, and $\mathbf{x}^{/d}$ is the sequence whose $d$-th component is removed or masked.

For positions with low confidence scores, the corrector step performs token-level resampling according to

$$\mathbf{x}_d \sim p_\theta(\cdot^{,d} \mid \mathbf{x}^{/d}).$$

Unlike GIDD and SCDD, Informed Corrector enables self-correction only at the sampling or inference stage. The self-correction mechanism itself is not learned during training, but instead relies on model-dependent confidence estimates at inference time.

**Plug-in Remasking for Inference-time Self-correction of Masked Diffusions (PRISM) (Kim et al., 2025)**   Given a pretrained masked diffusion model $p_\theta$, PRISM trains a token-level score function $g_\phi$ via

$$\mathcal{L}(\phi) := \mathbb{E}_{\mathbf{x},\mathbf{z},i,\mathbf{y} \sim p_\theta^i(\cdot|\mathbf{z})}\Big[\mathrm{CE}\big(\mathbf{1}[\mathbf{x}_i = \mathbf{y}_i], g_\phi^i(\mathbf{y})\big)\Big],$$

where CE denotes the cross-entropy loss. Here $\mathbf{x} \sim p_{\mathrm{data}}$ denotes the original clean sequence, $\mathbf{z} \sim q(\mathbf{z} \mid \mathbf{x})$ is obtained by the forward masking process in MDLM, and $\mathbf{y}^i \sim p_\theta^i(\cdot \mid \mathbf{z})$ is a token sampled from the pretrained MDM at a masked position $i$.

Intuitively, the score model $g_\phi^i(\mathbf{y})$ estimates the conditional probability

$$g_\phi^{i*}(\mathbf{y}) = \mathbb{P}(\mathbf{x}_i = \mathbf{y}_i \mid \mathbf{y} \oplus \mathbf{m}_i),$$

where the randomness is induced by the joint distribution over $\mathbf{x} \sim p_{\mathrm{data}}$, $\mathbf{z} \sim q(\mathbf{z} \mid \mathbf{x})$, and $\mathbf{y}^i \sim p_\theta^i(\cdot \mid \mathbf{z})$. Here $\mathbf{y} \oplus \mathbf{m}_i$ denotes the sequence obtained by masking the $i$-th token of $\mathbf{y}$, indicating that $\mathbf{y}_i$ is unobserved when evaluating $g_\phi^i$.

During inference, at each intermediate state $\mathbf{z}_t$, the learned score $g_\phi(\mathbf{z}_t)$ is used to identify unmasked tokens with low estimated quality. Such tokens are remarked by setting $\mathbf{z}_{t,d} = \mathbf{m}$, enabling inference-time self-correction through iterative remaking and unmasking.

**Path Planning Self-Planning (P2-Self)** ([Peng et al., 2025](#)) In the training stage, the authors follow the Masked Diffusion Language Model (MDLM) framework and train a denoising model

$$D_\theta : \mathcal{V}^L \to (\Delta^d)^L. \tag{61}$$

In inference, let

$$\mathbf{y} \sim D_\theta(\mathbf{x}_t). \tag{62}$$

The authors introduce and train a planner

$$G_\phi : \mathcal{V}^L \times \mathcal{V}^L \to [0,1]^L, \qquad (\mathbf{x}_t, \mathbf{y}) \mapsto G_\phi(\mathbf{x}_t, \mathbf{y}), \tag{63}$$

where $G_\phi^j(\mathbf{x}_t, \mathbf{y})$ denotes the probability that the $j$-th coordinate should be (re)sampled. Given adjacent time steps $s < t$, the reverse update is defined by

$$i \sim \frac{G_\phi(\mathbf{x}_t, \mathbf{y})}{\sum_{j=1}^{L} G_\phi^j(\mathbf{x}_t, \mathbf{y})}, \tag{64}$$

and

$$q_{t,\theta}(\mathbf{x}_s^i \mid \mathbf{x}_t^i) = \begin{cases} \mathrm{Cat}\left( \mathbf{x}_s^i; \dfrac{(1-\alpha_{t-1})\mathbf{m} + (\alpha_{t-1} - \alpha_t)D_\theta^i(\mathbf{x}_t)}{1-\alpha_t} \right), & \mathbf{x}_t^i = \mathbf{m}, \\[2ex] \mathrm{Cat}\left( \mathbf{x}_s^i; \dfrac{\left((\alpha_t - 1)D_\theta^i(\mathbf{x}_t)^\top \mathbf{x}_t^i + 1 - \alpha_{t-1}\right)\mathbf{x}_t^i + (\alpha_{t-1} - \alpha_t)D_\theta^i(\mathbf{x}_t)}{(1-\alpha_t)\left(1 - D_\theta^i(\mathbf{x}_t)^\top \mathbf{x}_t^i\right)} \right), & \mathbf{x}_t^i \neq \mathbf{m}. \end{cases} \tag{65}$$

Intuitively, $G_\phi(\mathbf{x}_t, \mathbf{y})$ selects a single coordinate $i$ to update. If $\mathbf{x}_t^i = \mathbf{m}$, the transition reduces to the original unmasking rule in MDLM. If $\mathbf{x}_t^i \neq \mathbf{m}$, the model remasks and resamples from $D_\theta$ at coordinate $i$ with some probability.

Similar to PRISM and Informed Corrector, this method does not incorporate self-correction in the pre-training stage. In addition, there is inherent redundancy in the inference stage: If $\mathbf{x}_{t,i}$ is masked, the resampling mechanism will not work.

# G. Unconditional Samples

In this section, we present unconditional samples generated by SCDD ($p_u = 0.2$) on OWT.

**T=32** ...years, we have not seen an case where a human computer can read a wall, the size of a human brain. It is enough to make a perfectly accurate world map, if you are able to create a digital map, with the map information.

The cameras are showing what they call "will", and how it will use it in the coming U of Indeed.e say the large number of Poles and soldiers expelled from Europe for being against the gold policy are going to do so, if will will be enough to them in their homes surrounded a themselves on the sea floor.

For the same reason they they, and right, not.

They told the vast canvass of research on the war is counter to the work of NATO's Chief Chiefsecurity, the Restipolar Prosperity Institute.

They describe a room "deposition room" that has a room room of 1,000 feet', eyes to 1,000 flying words. The cameras in the same room away from the listening table of the central room. The teleponics are high telescopes and small spectrums, out to the walls, the "theerences" can be seen as points of reference.

The cameras, they say, are't looking at people, of which the the information is stored, and are according to them, which no panoramic angles in order for information to settle in the small, confined space.

They say that the methods of stalking and shooting at targets will be obsolete, too. This works wonders in the military. One Facebookmanulated me that "in the country, it's possible that an man will shoot them if they have his gun" and that cells are used in in our forces.

They all began with arguments from young but gay, white anonymous authors. Professoricby said that the DUD can show that the man is, perhaps, "precompanceiveness of self-interest." They made a study that is "that they are just themselves victims are first first who be them," and that the are, for is are is come is the single target, but from what they are is do to them. However, they also offered a bias of the, such as "the difficulty of writing a novel with multiple readers," a novel about a young man living from home. And with the "mutual," the book will be...

**T=128** ...year, there will be about $100 billion of $1.50 wiped out by the dollar. The problem is that oil worth worth2 on the dollar and all of that that is there is banked by corruption that has a legal game to exist. And then there is the whole crisis that is, and the power of money and power. ...

Take the financial addiction. Over the past two years, drug banks and the Justice Department have have collapsed at rates far, many times the world oil price. When we did it last year, we had back in it. We should be helping money on law money. Why not raise the taxes on crime because of money? We fail, we are out of the moral era of the big-money cono.

But, it is not having the position to act a defendant in this case, said Sanjay Abdeh, the legal counsel at Steven Sinofsky, the chief in the case. Buffalo's law is the first to allow for marijuana marijuana for anyone who wants to buy marijuana. It hasn't been tested yet but Attorney General John Hentry said anyone who buys marijuana and wants it is legally selling the market. But since Florida's existing marijuana possession has required background checks, the Attorney General said it would be able to allow in legal medical marijuana into the state.

After the initial approval application for it last year, Gov. Rick Scott signed down a new law, allowing people able to possess the drug.

The state is moving away the a laws that would allow the use of the legally Florida along with others of state circumstances. Under the new law, the for marijuana won't be allowed to marijuana or of criminal possession, so it means state- criminal background check if the defense to be a subject able will turn that to possession the Justice Department will do.

The State Chief Information Officer John Betst said he is looking into the legal details as far as the amount and types what what is required and charges.

Advertisement

Florida's marijuana marijuana owner, Michelle Ibel, in Browesville County to meet and protect her husband and her three-year-old son. Ibel, a woman who lives from a war zone that has filled of violence, is believed to be one of the first people to be body-connell shot at a marijuana grow and medical center just off the street outside of the city's long-famous South City. Gang members are coming

out and about.  It is a new war...

**T=1024**  ...on the different screen, that new features are going to come out, and the car is at least going of high promise.  Stay tuned as we look forward to the future.

It's clear from the images that the product is much faster is it from Tesla.  It will be rolling in two months of deliveries from now to mid-year.  Below you can see the deliveries align into each other.

When product is unveiled at E3 of this year it will be only (only) in the car's first few months.

By all accounts, the company's product generation of products is well under the wings.  Menting new companies are butnecessarily, products that have the attention of other players.  It is also in into that major players not, such as the Apple ecosystem.

On its side, Tesla could be a car car company.  While Tesla has already worked with Uber and Lyft, it is going with the automaker with the direction creating an app where consumers will use better cars.

The company is going down the technology in a new direction.

"As Tesla brings together the same strengths and technology of the Prius software company, the result is better and better cars, more, electric cars, better performance on the road and better car safety and data."

Tesla has seen since the with competitive advances, but those intangibles have, and last year its margins were even better."[Source]

Via Bloomberg
Donald Trump has been vocal about concerns about the "radical and media agenda" since to 9/11.

But, that doesn't mean Trump is going to address his concerns about the Trump Trump.  It has been a lot of debate.

In January, Donald Trump said, "I's not to be great to you have to interviews in a pieces, a president's opinion on these different issues" and made it a point that "the media word-for-word on the story I put out."

Donna Trump asked a similar opinion.  She asked Trump, "doesn't Trump sound like a great leader of the country?" And his "strategy is winning, and he tells the truth understand." It's a big question as the calls for the right to run for president.  Over the past week, she has been much debate about Trump's...

