# OpenReview forum: "Generalized Discrete Diffusion with Self-Correction"
_ICML.cc/2026/Conference — ICML 2026 regular_

### Official Review · Reviewer_PMQ4 · 2026-03-05

**Soundness:** 3
**Presentation:** 3
**Significance:** 3
**Originality:** 3
**Overall Recommendation:** 5
**Confidence:** 4

**Summary:**

This paper proposes a new framework self-correcting discrete diffusion (SCDD) for masked diffusion language models (MDLMs). Standard MDLMs lack the ability to modify its tokens once they are unmasked. While self-correction can improve this, previous methods often rely on post-training or complicated inference-time sampling techniques.

In previous work GIDD, the uniform state transition is coupled to the masking probability. SCDD reformulates the pretraining-based self-correction process by introducing a clear and explicit state transition in discrete time. The key innovation is the separation of uniform transitions and mask absorbing, which allows individual noise schedules for independent control.

In the experiments, SCDD achieves better generative perplexity on GPT-$2$ SMALL scale experiments. The paper also investigates the impact of the noise schedule to the self-correcting ability of the model. It shows that setting a larger uniform noise ratio allows a more aggressive parallel self-correction of the model. In addition, by shifting the timing of peak uniform noise to the end, the model is able to complete almost 40% of the correction within the first $200$ steps.

**Compliance With Llm Reviewing Policy:**

Affirmed.

**Final Justification:**

My concerns of this work is addressed by the authors in the rebuttal and I maintain my positive assessment of this work.

**Key Questions For Authors:**

1 . While matching the GIDD’s noise schedule provides a solid baseline for comparison, this leaves the optimal schedule for SCDD unexplored. Do the authors have preliminary results on making the noise schedule parameters learnable during training?

2. The experiments show that increasing $p_u$ increases correction rates but also degrades validation perplexity. Do the authors believe that there is a ‘sweep spot’ for the noise schedule optimizing that maximizes generation efficiency without sacrificing reasoning capabilities?

3. Current experiments are conducted using SMALL architecture. Do the authors anticipate any shifts in optimal noise schedules, or changes in self-correction behaviour when scaling?

4. Could the authors explain why the enhanced self-correction capability negatively impacts zero-shot likelihood as indicated in Table 6?

**Limitations:**

Yes

**Strengths And Weaknesses:**

Strength:
 - The mathematical formulation is clean and offers a significant improvement over previous methods in the literature, such as GIDD. In GIDD, the uniform transition noise and mask noise are coupled. By reformulating the forward process allowing independent control over uniform and mask transition rates, this work provides a more interpretable framework than previous continuous-time formulations.
 - Since SCDD is trained on unifrom transition of unmased tokens, it is able to correct unmasked token in one step compared to the purely remask-based methods. This translates to clear empirical evidence: The experiments show a clear improvement in generative perplexity (9.2% decrease) compared to other remasking-based self-correcting methods.
- The paper provides a thorough ablation study investigating the impace of noise schedules. By modifying the max uniform ratio $p_u$ and the timing of peak noise $t_{\text{peak}}$, this gives a clearer understanding of the mechanism and the model behaviors.

Weakness:
 - The validity of the method is verified using a DiT backbone at the SMALL scale. Given the current trend of large language models, it is unclear if the observed self-correcting mechanics is able to transfer to larger models.
 - Introducing additional uniform transition enabled self-correction while degrades the performance of the model on general reasoning benchmarks compared to standard mask-only MDLMs. This is commonly observed both in SCDD and previous work GIDD. The paper notes that these benchmarks primarily measure zero-show likelihood rather than self-correction ability. However, this decrease in performance is still a notable limitation to the methods.

---

> ### Author Rebuttal · Authors · 2026-03-31
>
> ## W1
> > The validity of the method...
>
> **Answer**: Regarding SCDD at a larger scale (e.g., LLaDA), we refer to the answer to "W4" from reviewer "nm67" for details.
>
> ## W2
> > Introducing additional uniform transition...
>
> **Answer**:  Regarding the performance of MDLM vs. GIDD+/SCDD, we refer to the answer to "Q3" from reviewer "fyya" for details.
>
> ## Q1
> > While matching the...
>
> **Answer**: We thank the reviewer for this suggestion. We agree that the optimal noise schedule for the mixed mask/uniform corruption setting remains underexplored. In our main experiments, we adopt the same schedule as GIDD to ensure a fair comparison.
>
> Theoretically, making $(\rho_t, \gamma_t)$ learnable is nontrivial. It requires constrained parameterization (e.g., monotonicity) and likely additional regularization to avoid degenerate schedules. Moreover, since $(\rho_t, \gamma_t)$ affect both the integrand and the reference measure in $\mathcal{L}^T$ and $\mathcal{L}^\infty$, jointly optimizing them with the model introduces additional non-convexity. Therefore, it is unclear what objective and constraints are most appropriate without changing the original problem setting.
>
> In practice, we conducted a preliminary experiment on WikiText-103 using the monotonic polynomial parameterization from Sahoo et al. [1]. In particular, we set
> $$
> \gamma(t) = e^{-f(t)}, \qquad \rho(t) = e^{-g(t)},
> $$
>
> where $f$ and $g$ are monotonic polynomials, with bounds chosen so that the marginal distribution is well-defined.
> However, we observe a degenerate solution: as training proceeds, $\gamma(t)$ collapses toward zero for nearly all $t$. One possible reason is that, in our loss (Eqs. (12) and (13)), letting $\gamma_t \to 0$ reduces the contribution of Case 2, while also lowering the probability that a token remains unmasked in Case 1. As a result, the optimization is driven toward a trivial schedule.
>
> Overall, these preliminary results suggest that directly learning the noise schedule is challenging and likely requires a better-designed objective or additional constraints. We therefore leave this direction to future work. We refer to `/rebuttal/schedule_learning/` in https://anonymous.4open.science/r/SCDD_Rebuttal_Update for reproduction.
>
> ## Q2
> > The experiments show...
>
> **Answer:**
> Yes, we believe that there is a "sweet spot." This is expected because increasing $p_u$ strengthens the model's self-correction ability by exposing it to more uniformly corrupted states, but excessively large $p_u$ may overly disrupt the underlying data structure and hurt likelihood modeling and general reasoning performance, given constrained model capacity and computational resource. Therefore, the optimal noise schedule should balance these two effects: sufficient corruption to encourage correction, but not so much that it degrades the model's predictive quality.
>
> ## Q3
> > Current experiments...
>
> **Answer**: We refer to the answer to "W4" from reviewer "nm67" for details.
>
> ## Q4
> > Could the authors explain...
>
> **Answer**:
>
> We acknowledge the lower zero-shot scores and appreciate the reviewer raising this point. However, each of zero-shot tasks ranks answer candidates by their ELBO so that there is no self-correction during evaluation, making these benchmarks a measure of zero-shot likelihood rather than language generation. Uniform-noise diffusion models (including both SCDD and GIDD+) typically exhibit worse likelihood but stronger abilities in self-correction, few-step sampling, and generation tasks [2]. Our generative PPL results (Table 3) confirm that SCDD produces higher-quality text than MDLM/GIDD at equal sampling steps, demonstrating that worse likelihood estimation does not translate to weaker language generation.
>
> ## Reference
> [1] Sahoo, Subham S., et al. "Diffusion models with learned adaptive noise." Advances in Neural Information Processing Systems 37 (2024): 105730-105779.
>
> [2] Sahoo, Subham Sekhar, et al. "Scaling Beyond Masked Diffusion Language Models." arXiv preprint arXiv:2602.15014 (2026).

---

> > ### Author Rebuttal · Reviewer_PMQ4 · 2026-04-03
> >
> > Thanks to the authors for clarification and the addtional experiments.

---

### Official Review · Reviewer_nm67 · 2026-03-05

**Soundness:** 2
**Presentation:** 3
**Significance:** 2
**Originality:** 2
**Overall Recommendation:** 4
**Confidence:** 3

**Summary:**

This paper proposes Self-Correcting Discrete Diffusion (SCDD) to enable self-correction in discrete diffusion language models. The authors reformulates pretrained self-correction with explicit discrete-time state transitions, simplifies the noise schedule, removes redundant remasking steps, and uses uniform transitions to learn self-correction.

**Compliance With Llm Reviewing Policy:**

Affirmed.

**Final Justification:**

The rebuttal has addressed most of my concerns, so I will give the paper a positive score.

**Key Questions For Authors:**

1. Please see the weakness part.
2. Since the method focuses on the pretraining phase, is it possible to apply the training target during post-training?

**Limitations:**

No. The paper focuses only on small-scale models, and in some cases its performance is still less satisfactory than MDLM.

**Strengths And Weaknesses:**

Strengths
1. The motivation is clear. The paper focuses on improving parallel generation via self-correction without adding complex sampling strategies.
2. The proposed method introduces no remasking, which could potentially reduce inference latency.
3. The presentation is relatively clear, providing comprehensive formulations and proofs.

Weaknesses
1. The empirical results do not support the claimed advantage. In Table 1, SCDD is clearly worse than MDLM, which is a stronger and more relevant baseline than GIDD. This weakens the practical impact of the method.
2. The paper contends improved efficiency for parallel decoding, but does not provide any direct speed evidence. There are no latency, throughput, or FLOPs comparisons, so the efficiency claim is not sufficiently validated.
3. The evaluation is limited. The paper mainly relies on perplexity, entropy, and its own correction-related metrics, but does not provide stronger semantic or human evaluation to show that the outputs are actually better.
4. The experiments are narrow. The method is only tested on small-scale models, and there is no validation on larger diffusion LLMs such as LLaDA or Dream, so it is unclear whether the method works at scale.
5. It is also unclear whether the “corrections” are actually beneficial. The paper measures how often tokens are changed, but does not analyze whether these changes truly fix incorrect outputs, but not making meaningless edits or same-meaning substitutions. Therefore, the self-correction ability is not fully validated.

---

> ### Author Rebuttal · Authors · 2026-03-31
>
> ## W1:
> > The empirical results...
>
> **Answer**: We refer to the answer to Q3 in reviewer "fyya".
>
> ## W2:
> > The paper contends...
>
> **Answer**: We thank the reviewer for the comment. First, we clarify that the efficiency of our method refers to the fact that, during the generation stage, the new method does not perform a remasking step for self-correction. Thus, the new method requires fewer sampling steps (or NFEs) to reach the same performance. We refer to Tables 3 in the original submission for details. Theoretically, it is shown that diffusion LLMs are optimal parallel samplers [1].
>
> ## W3:
> > The evaluation is...
>
> **Answer**:
> We add an additional **LLM-as-a-judge** evaluation to directly assess correction quality beyond perplexity, entropy, and our correction-related metrics.
>
> For each experiment, we corrupt 256 clean OpenWebText sequences from $t=0$ to $t=0.8$ using SCDD's forward process, then let both models denoise from the identical corrupted input. Each model generates outputs at 6 step counts (32, 64, 128, 256, 512, 1024) with nucleus sampling ($p=0.9$). The GPT-5.4 judge scores each pair on five dimensions (clarity, grammaticality, factuality, style, creativity) on a 1--10 scale without knowing which model produced which output. Finally, the model decides the "winning text". We report per-metric mean differences with paired $t$-tests and overall win rates with binomial tests ($n=256$). The evaluation prompt for GPT-5.4 strictly follows Figure 7 in GIDD paper.
>
> **Table 1: Matched schdule setting** — SCDD ($p_u=0.2$) vs GIDD+ ($p_u=0.2$), 1024 steps
>
> | Dimension | SCDD | GIDD+ | Diff | SE | 95% CI | $p$ (paired $t$-test) | Cohen's $d$ |
> |---|:---:|:---:|:---:|:---:|:---:|:---:|:---:|
> | Clarity | 1.73 | 1.48 | +0.25 | 0.054 | [+0.14, +0.36] | <0.001\*\*\* | +0.45 |
> | Grammaticality | 1.57 | 1.46 | +0.11 | 0.054 | [+0.01, +0.22] | 0.036\* | +0.21 |
> | Factuality | 2.20 | 2.07 | +0.13 | 0.043 | [+0.04, +0.21] | 0.004\*\* | +0.11 |
> | Style | 1.73 | 1.50 | +0.24 | 0.054 | [+0.13, +0.35] | <0.001\*\*\* | +0.43 |
> | Creativity | 2.93 | 3.15 | −0.22 | 0.049 | [−0.32, −0.13] | <0.001\*\*\* | −0.29 |
> | **Win rate** | | | **60.6%** | | | **0.001\*\*** | |
>
> **Table 2: Cross-ratio setting** — SCDD ($p_u=0.2$) vs GIDD+ ($p_u=0.1$), 1024 steps
>
> | Dimension | SCDD | GIDD+ | Diff | SE | 95% CI | $p$ (paired $t$-test) | Cohen's $d$ |
> |---|:---:|:---:|:---:|:---:|:---:|:---:|:---:|
> | Clarity | 1.73 | 1.52 | +0.22 | 0.057 | [+0.10, +0.33] | <0.001\*\*\* | +0.37 |
> | Grammaticality | 1.51 | 1.49 | +0.02 | 0.053 | [−0.09, +0.12] | 0.712 | +0.04 |
> | Factuality | 2.12 | 2.01 | +0.11 | 0.047 | [+0.02, +0.21] | 0.017\* | +0.10 |
> | Style | 1.72 | 1.52 | +0.19 | 0.057 | [+0.08, +0.30] | <0.001\*\*\* | +0.33 |
> | Creativity | 2.98 | 3.27 | −0.29 | 0.049 | [−0.38, −0.19] | <0.001\*\*\* | −0.35 |
> | **Win rate** | | | **58.0%** | | | **0.014\*** | |
>
> Significance levels: * $p < 0.05$, ** $p < 0.01$, *** $p < 0.001$ (two-sided paired $t$-test).
>
> We leave the remaining tables to `rebuttal/llm_as_a_judge/tables_all_steps.md` in https://anonymous.4open.science/r/SCDD_Rebuttal_Update. SCDD is consistently the most capable model of handling noisy texts.
>
> ## W4
> > The experiments are...
>
> **Answer**: We agree that training at larger scale is an important next step. Due to the rebuttal timeline and computational constraints, we are currently unable to pretrain a model at the LLaDA's scale. That said, recent large-scale diffusion LLMs (e.g., LLaDA-2.1 [2]) have already suggested that pretraining or post-training strategies related to self-correction / iterative refinement can be beneficial, which supports the relevance of this direction beyond the small-scale setting. Since our contribution is a training objective rather than an architecture-specific modification, we believe it is in principle compatible with larger diffusion LLMs as well, and we leave it to future work.
>
> ## W5
> > It is also unclear...
>
> **Answer**:  We acknowledge that it is difficult to track all corrections and evaluate whether they are truly beneficial. While the additional LLM-as-a-judge experiments in W3 do not provide a perfect evaluation of correction success, we believe they offer indirect evidence of correction quality by assessing whether the additional revisions made by SCDD tend to improve the generated text, rather than merely increase the number of edits.
>
> # Q2.
> > Since the method focuses ...
>
> **Answer**: We believe it is possible. For example, we can perform SFT with LLaDA base models using the SCDD loss function, avoiding computation expenses and learn how to self-correct in a broader context. We refer the reviewer to our response to W4 for more details.
>
> ## Reference
>
> [1] Jiang, Haozhe, Nika Haghtalab, and Lijie Chen. "Diffusion Language Models are Provably Optimal Parallel Samplers." arXiv preprint arXiv:2512.25014 (2025).
>
> [2] Bie, Tiwei, et al. "Llada2. 1: Speeding up text diffusion via token editing." arXiv preprint arXiv:2602.08676 (2026).

---

> > ### Author Rebuttal · Reviewer_nm67 · 2026-04-03
> >
> > Thank you for the rebuttal.
> >
> > I would like to further clarify my concern on efficiency. In the paper, Table 2 compares the performance of different methods under the same number of denoising steps, which is not the comparison I had in mind. My question is about the actual generation cost for solving the same problem: for the same input, how much latency time does each method require? The current rebuttal mainly argues that SCDD may need fewer sampling steps because it avoids remasking, but this still does not provide direct evidence on latency or throughput.
> >
> > For W5, although the added LLM-as-a-judge evaluation is helpful, it only provides indirect evidence. The paper does not directly verify whether the “corrections” are beneficial edits rather than unnecessary changes. Therefore, I still view this as a weakness of the current submission.

---

> > > ### Author Response · Authors · 2026-04-05
> > >
> > > Thank you very much for clarifying the questions.
> > > ## Latency
> > > For the latency concern, since both SCDD and GIDD+ perform a single backbone forward pass per denoising step and differ only in the backward process, the two methods have the same per-step computational complexity in principle. To verify this empirically, we conducted a wall-clock benchmark. To ensure fair comparison, we applied `torch.compile` to the SCDD sampler to match the compiled `DenoisingStep` already present in GIDD+. Both methods were then benchmarked on the same A100 GPU with batch size 16, sequence length 512, 3 warmup runs, and 10 timed runs.
> > >
> > > We measure two metrics:
> > > - **Latency**: Total wall-clock time to generate one batch of 16 sequences of length 512.
> > > - **Per-step time**: Latency / number of denoising steps.
> > >
> > > | Steps | SCDD Latency (s) | GIDD+ Latency (s) | SCDD Per-Step (ms) | GIDD+ Per-Step (ms) | Speedup |
> > > |:---:|:---:|:---:|:---:|:---:|:---:|
> > > | 32 | 0.931 | 1.550 | 29.1 | 48.4 | 1.67x |
> > > | 64 | 1.862 | 3.097 | 29.1 | 48.4 | 1.66x |
> > > | 128 | 3.730 | 6.204 | 29.1 | 48.5 | 1.66x |
> > > | 256 | 7.496 | 12.429 | 29.3 | 48.6 | 1.66x |
> > > | 512 | 15.106 | 24.874 | 29.5 | 48.6 | 1.65x |
> > >
> > > The results show a consistent ~1.66x per-step speedup for SCDD. However, we note this gap is implementation-dependent — it likely reflects differences in how `torch.compile` fuses each method's backward process rather than a fundamental algorithmic advantage. We do not claim per-step speed as a contribution of SCDD. What matters more is few-step generation quality. As shown in Table 2 of the paper, SCDD achieves better perplexity than GIDD+ at the same step count, meaning it needs fewer steps to reach a given quality target.
> > >
> > > ## W5
> > > We note that evaluating whether each individual correction is beneficial is inherently difficult, as intermediate steps contain many mask tokens that obscure the effect of any single edit. Nevertheless, we designed two experiments that try to quantify the quality of corrections as directly as possible.
> > >
> > > **Experiment 1: Ablation.** We ran SCDD with corrections disabled by freezing all previously unmasked tokens at each step, so only masked positions can change. Note that this deviates from the theoretical sampler, which allows all positions to be updated; everything else remains identical (same model weights, noise schedule, 128 denoising steps, nucleus $p = 0.9$). We generated 256 sequences of length 512 and evaluated Gen-PPL under GPT-2-large.
> > >
> > > | Variant | Gen-PPL (GPT-2-large) $\downarrow$ |
> > > |:---|:---:|
> > > | SCDD (with corrections) | 59.17 |
> > > | SCDD (no corrections) | 188.13 |
> > >
> > > Disabling corrections degrades perplexity from 59.17 to 188.13. Since the only variable changed is whether the model may revise previously generated tokens, this provides direct causal evidence that corrections are essential for generation quality.
> > >
> > > **Experiment 2: Corruption Recovery.** To provide more direct evidence that corrections fix errors at the token level, we took clean validation text, randomly corrupted $K$ tokens per sequence with uniform random replacements, and ran a single SCDD denoising step at the last-step noise level ($t \approx 0.008$, corresponding to step 127 of 128). We report two metrics: **touch rate**, the fraction of corrupted positions where the model produces a different token than the corrupted one, and **recovery rate**, the fraction of corrupted positions where the model restores the exact original clean token. Results are averaged over 128 sequences with standard errors.
> > >
> > > | $K$ | Touch Rate | Recovery Rate | PPL (corrupted) | PPL (corrected) |
> > > |:---:|:---:|:---:|:---:|:---:|
> > > | 5 | 1.000 $\pm$ 0.000 | 0.694 $\pm$ 0.013 | 22.0 $\pm$ 0.3 | 23.8 $\pm$ 0.5 |
> > > | 10 | 1.000 $\pm$ 0.000 | 0.652 $\pm$ 0.011 | 28.2 $\pm$ 0.4 | 24.0 $\pm$ 0.4 |
> > > | 20 | 0.999 $\pm$ 0.001 | 0.647 $\pm$ 0.008 | 44.7 $\pm$ 0.7 | 24.3 $\pm$ 0.5 |
> > > | 50 | 1.000 $\pm$ 0.000 | 0.644 $\pm$ 0.005 | 154.8 $\pm$ 2.2 | 25.5 $\pm$ 0.4 |
> > >
> > > The model detects nearly 100% of corrupted tokens and **exactly recovers 64–69%** of them. The perplexity improvements are substantial: at $K = 50$ (~10% corruption), perplexity drops from 154.8 to 25.5, recovering most of the way to the clean baseline of 16.9. We noticed that the model also modifies some uncorrupted tokens, but these changes are mainly semantic paraphrases rather than errors (e.g., "can be viewed as an attempt" $\to$ "could be seen as an opportunity" in the first example in `examples_k50.json`), consistent with the corrected PPL staying close to baseline.
> > >
> > > The code for reproduction are updated in https://anonymous.4open.science/r/SCDD_Rebuttal2. More last-step correction examples are stored in `/rebuttal/correction/examples_k50.json`.

---

### Official Review · Reviewer_fyya · 2026-03-14

**Soundness:** 3
**Presentation:** 3
**Significance:** 2
**Originality:** 3
**Overall Recommendation:** 4
**Confidence:** 3

**Summary:**

This paper proposes self-correction discrete diffusion (SCDD), a reformulation of pretraining based self-correction for discrete diffusion language models. Building on prior work such as GIDD, the method introduces an explicit discrete-time forward process with decoupled masking and uniform corruption schedules, together with an absorbing state that removes remasking during reverse denoising. The authors derive the corresponding reverse posterior and ELBO objective, and evaluate the method at GPT-2 scale on LM1B and OpenWebText.

**Compliance With Llm Reviewing Policy:**

Affirmed.

**Final Justification:**

The rebuttal addressed my questions and concerns.

**Key Questions For Authors:**

1. Could the author explain the performance difference between the reported PPL for GIDD versus the original paper reported in Table 3?
2. Does “higher correction capacity” reflect successful self-correction, or only more frequent token revisions? It would help to clarify whether the corrections are actually improving the sample rather than simply increasing edit activity.
3. Why do models trained on uniform and mask noises underperform the mask-only models? I assume your argument is that the former is a more difficult task but to me, adding self-correction provides more flexibility to the model. Are we limited by model capacity? This brings back to why do we want to have self-correction of a discrete diffusion models in the first place given we can't get better likelihood.

**Limitations:**

yes

**Strengths And Weaknesses:**

**Strengths**
- The math in clean in section 3 and the derivations in the main text are easy to follow.
- With the novel design of the forward process and the absorbing state formulation, the proposed model achieves correction without remasking for more efficient self-correction.
- I think the paper studies an interesting and timely problem. Improving self-correction in discrete diffusion language models is a meaningful direction, and progress in this area could have substantial impact on parallel text generation.

**Weakness**
- The main concern I have is in the experiment section,  the baseline performance reported in Table 1 for GIDD is far from what is being reported in the original paper given the same model size and the same number of training tokens. This also makes reviewer difficult to judge the model's performance Table 3. Please see questions section for more details.

---

> ### Author Rebuttal · Authors · 2026-03-31
>
> ## W1 & Q1
> > The main concern I have...
>
> > Could the author ...
>
> **Answer**:
> We refer the reviewer to our answer to “Soundness 2” from reviewer FXK8 for details.
>
> ## Q2.
> > Does “higher correction capacity”...
>
> **Answer**: The phrase **“higher correction capacity”** in our paper does not mean that we explicitly verify whether each revision is a successful correction in a strict semantic sense. Rather, our point is that the proposed formulation removes the redundant remasking step during correction, thereby creating more opportunities for revision within the same generation process.
>
> At the same time, we agree that the quality of these revisions is important. Since “successful self-correction” is difficult to define and measure precisely, we conducted two additional **LLM-as-a-judge** experiments using the **GPT-5.4 API** to compare SCDD with GIDD. In these experiments, the judge evaluates the generated text along five dimensions: **clarity, grammaticality, factual consistency, style, and creativity**. While these experiments do not provide a perfect definition of correction success, they offer indirect evidence of correction quality by assessing whether the additional revisions made by SCDD tend to improve the generated text rather than merely increase the number of edits. We refer the reviewer to our response to **Reviewer nm67** W3 for the detailed settings and results.
>
> ### Q3.
> >Why do ...
>
> **Answer**: Our view is that training with both uniform and mask noise introduces a strictly harder denoising problem than the mask-only setting: the model must recover from a broader range of corrupted states, rather than from masking alone. Although this added flexibility can be beneficial at generation time, it also increases optimization difficulty under the same model capacity and training budget, which is why likelihood does not necessarily improve over mask-only diffusion LLMs.
>
> More importantly, likelihood estimation and generation quality are not always perfectly aligned empirically [1][2]. The purpose of self-correction in discrete diffusion models is not solely to maximize likelihood, but to enable the model to learn how to revise its own predictions during pretraining stage, thereby improving few-step parallel decoding ability for faster generation. In mask-only models, the generation process is relatively rigid because once a token is revealed, it is typically fixed thereafter. By contrast, self-correction allows the model to revisit earlier predictions, which is especially valuable when many tokens are generated in parallel and early mistakes can propagate.
>
> ## Reference
> [1] Von Rütte, Dimitri, et al. "Generalized interpolating discrete diffusion." arXiv preprint arXiv:2503.04482 (2025).
>
> [2] Sahoo, Subham Sekhar, et al. "Scaling Beyond Masked Diffusion Language Models." arXiv preprint arXiv:2602.15014 (2026).

---

> > ### Author Rebuttal · Reviewer_fyya · 2026-04-04
> >
> > Thank your for your answers to my questions especially on W1 and Q2 and Q3.  I will raise my score given the response.

---

### Official Review · Reviewer_FXK8 · 2026-03-15

**Soundness:** 2
**Presentation:** 4
**Significance:** 2
**Originality:** 4
**Overall Recommendation:** 4
**Confidence:** 5

**Summary:**

The authors propose SCDD, a self correction approach for discrete diffusion models, without the need for remasking steps, explicitly introducing controllable uniform transition steps to learn self-correction. The authors conduct experiments at GPT2 scale, on LM1B and OWT data, demonstrating reasonable quality while depicting higher self correction rate than the other significant baseline, GIDD.

**Compliance With Llm Reviewing Policy:**

Affirmed.

**Final Justification:**

mostly concerns are addressed. weak accept because of the weak empirical results overall. i like the paper theoretically.

**Key Questions For Authors:**

Covered in the sections above

**Limitations:**

The authors briefly discuss the weak empiricial performance of their approach, but I believe that these aspects require further work/justification as suggested above

**Strengths And Weaknesses:**

1. Soundness:
- The paper is extremely well written from a theoretical perspective, with clear motivation, elegant formulation and detailed proofs. I found a couple of minor issues which I'll flag here. (a) The closed form expression derivation for $\bar{A_t}$ should use the index j instead of i. (b) The first case of the backward denoising process on line 205 ($z_t \neq m$) implies that ($z_s \neq m$) as pointed in the proof as well. So, the multiplicative term $(1-z_s^Tm)$ is guaranteed to be $1$, and need not be added. Even when $x$ is parameterized, having this term doesn't help at all.
- My primary critique of the paper comes from its empirical evaluation. In Table 1, the GIDD baseline is retrained by the authors to have a fair empirical evalution, by setting the GIDD params in compliance with SCDD params, so that both methods see distributionally equivalent samples during training. However, the original GIDD paper reports a OWT pplx of 23.67 with $p_u=0.1$ (as compared to author reported 31.54), and 24.38 with $p_u=0.2$ (vs author reported 32.19). A framework equivalence argument does not justify a 8 point perplexity gap at GPT2 scale, and significantly weakens SCDD's empirical positioning.
- The validation perplexity increase in SCDD as compared to MDLM is not well explained, and left at "higher difficulty of learning transitions between non [MASK] tokens". But, the initial motivation for this framework was having decoupled uniform and mask transitions for easier learning. This needs further justification. Additionally, Table 6 shows lower zero shot performance for SCDD than MDLM. The argument that these benchmarks dont measure self correction is unconvincing here. This seems to be an indicator of pooer language modeling capabilities with SCDD. Additionally, Table 5 shows that SCDD has lower entropy than GIDD and MDLM, which might mean lower diversity generations, and weakens the generative perplexity gains shown in Table 3.

2. Presentation:
- The theoretical presentation is excellent (expect a couple issues pointed above), and the paper is easy to reproduce by reading the paper. The previous methods are appropriately discussed and compared against in the paper. The empirical section has reasonable evaluations (GPT2 scale trained) for a pretraining based approach like SCDD.

3. Significance:
- The theoretical elegance of the method is currently severely hindered by its weak evaluation results and seemingly unfair baselines. The authors should focus on clarifying these issues, to help position the paper better in the discrete diffusion space.

4. Original:
- There are a few self-correction approaches for discrete diffusion, but only GIDD does it during pretraining other than this paper. The approach followed in this paper is quite novel.

On an overall level, I am very supportive of the theoretical framework proposed, but it is critical for the authors to justify their empirical results.

---

> ### Author Rebuttal · Authors · 2026-03-31
>
> ## Soundness 1
> > I found a couple of minor...
>
> **Answer**:
>
> (a) We've fixed the notation typos in Lines 187--197.
> (b) We've removed $1 - z_s^\top \mathbf{m}$ in the first case of Eq. (8).
>
> ## Soundness 2
> > My primary critique...
>
> **Answer**:
>
> ## Validation perplexity computation in the experiments
>
> Diffusion LLMs typically estimate validation perplexity (Val PPL) through an approximation of the NELBO. As a result, the reported Val PPL depends on the specific evaluation protocol and can vary across implementations. To make the comparison precise, we distinguish three versions of GIDD Val PPL:
> 1. **v1**: the Val PPL reported in our initial submission, computed from our retrained GIDD checkpoints using the GIDD-based evaluation protocol.
> 2. **v2**: the Val PPL computed during rebuttal from the released GIDD checkpoints on Hugging Face, using the same GIDD-based evaluation protocol.
> 3. **v3**: the Val PPL reported in the original GIDD paper.
>
> We first compare **v2** and **v3**. Using the released GIDD checkpoints and the same GIDD-based protocol adopted in our paper, we obtain Val PPL values of **27.79** for $p_u=0.1$ and **28.71** for $p_u=0.2$, which are already notably higher than the **23.67** and **24.38** reported in the original GIDD paper. We believe this gap mainly comes from differences in evaluation protocol. Since diffusion-model Val PPL is computed from an NELBO-type estimate and then exponentiated, even small differences in the ELBO/NELBO approximation or sampling variance can lead to noticeable differences in the final Val PPL. Please see `rebuttal/val_ppl_calculation/` in https://anonymous.4open.science/r/SCDD_Rebuttal_Update for the code used to compute **v2**.
>
> We next compare **v1** and **v2**. A residual gap remains between our retrained GIDD checkpoints and the released checkpoints: **31.54 vs. 27.79** for $p_u=0.1$, and **32.19 vs. 28.71** for $p_u=0.2$. We believe this difference mainly comes from differences in the training setup and the resulting checkpoints. Although we matched the total number of 131B training tokens, our retraining uses a batch size of 256 for 1M steps, whereas the released checkpoints use a batch size of 512 for 500k steps. Therefore, the gap between **v1** and **v2** should primarily be attributed to training differences rather than the evaluation code itself.
>
> We will clarify these distinctions in the revision so that the comparison with GIDD is more precise and transparent.
>
> ## Soundness 3.
> > The validation perplexity increases...
>
> **Answer**:
> For the parameter decoupling question, we want to clarify that the "decouple of uniform and mask transition" doesn't make the learning task itself easier. Instead, it improves the interpretation of the noise schedule and helps to control the forward noising process and backward self-correction dynamics.
>
> For the zero-shot benchmark question, we refer the reviewer to our response to Reviewer PMQ4, Q4, for details.
>
> For the trade-off between entropy and generative perplexity, entropy degradation is also observed in Kim et al. [1]. We believe SCDD's entropy is still within a reasonable range and won't hurt the quality of the generated texts.
>
> ## Reference:
> [1] Kim, Jaeyeon, et al. "Fine-tuning masked diffusion for provable self-correction." arXiv preprint arXiv:2510.01384 (2025).

---

> > ### Author Rebuttal · Reviewer_FXK8 · 2026-04-04
> >
> > thank you for the rebuttal. i stilll believe that the empirically weak results undercut the significance of the results to some extent, however given the strong theoretical narrative and reasonably well presented results, im happy to move towards acceptance by one point.

---

### Decision · Program_Chairs · 2026-04-30

**Decision:**

Accept (regular)

**Comment:**

All three reviewers converged on weak accept or accept, satisfied by the rebuttal. The remaining concerns is the small-scale experiments, which was acknowledged by the authors as known limitations rather than fundamental flaws of the method. Given the increasing satisfaction by the reviewers during the rebuttal, this is recommended for acceptance.